# Rapid termination of the African Humid Period triggered by northern high-latitude cooling

James A. Collins [1,2,3], Matthias Prange[3], Thibaut Caley[4], Luis Gimeno[5], Britta Beckmann[3], Stefan Mulitza[3], Charlotte Skonieczny[6], Didier Roche [7,8] & Enno Schefuß[3]

The rapidity and synchrony of the African Humid Period (AHP) termination at around 5.5 ka are debated, and it is unclear what caused a rapid hydroclimate response. Here we analysed the hydrogen isotopic composition of sedimentary leaf-waxes ($\delta D_{wax}$) from the Gulf of Guinea, a proxy for regional precipitation in Cameroon and the central Sahel-Sahara. Our record indicates high precipitation during the AHP followed by a rapid decrease at 5.8–4.8 ka. The similarity with a $\delta D_{wax}$ record from northern East Africa suggests a large-scale atmospheric mechanism. We show that northern high- and mid-latitude cooling weakened the Tropical Easterly Jet and, through feedbacks, strengthened the African Easterly Jet. The associated decrease in precipitation triggered the AHP termination and combined with biogeophysical feedbacks to result in aridification. Our findings suggest that extratropical temperature changes, albeit smaller than during the glacial and deglacial, were important in triggering rapid African aridification during the Holocene.

[1] GFZ–German Research Center for Geosciences, Section 5.1 Geomorphology, Organic Surface Geochemistry Lab, D-14473 Potsdam, Germany. [2] AWI-Alfred Wegener Institute Helmholtz Centre for Polar and Marine Research, Am Alten Hafen 26, D-27568 Bremerhaven, Germany. [3] MARUM—Center for Marine Environmental Sciences, University of Bremen, D-28359 Bremen, Germany. [4] EPOC, CNRS, University of Bordeaux, Allée Geoffroy Saint-Hilaire, 33615 Pessac Cedex, France. [5] Environmental Physics Laboratory (EPhysLab), Facultade de Ciencias, Universidad de Vigo, 32004 Ourense, Spain. [6] Laboratoire GEOsciences Paris-Sud (GEOPS), UMR CNRS 8148, Université de Paris-Sud, Université Paris-Saclay, 91405 Orsay Cedex, France. [7] Faculty of Earth and Life Sciences, Earth and Climate Cluster, Vrije Universiteit Amsterdam, De Boelelaan 1085, 1081 HV Amsterdam, The Netherlands. [8] Laboratoire des Sciences du Climat et de l'Environnement (LSCE), CEA/CNRS-INSU/UVSQ, 91191 Gif-sur-Yvette Cedex, France. Correspondence and requests for materials should be addressed to J.A.C. (email: jcollins@gfz-potsdam.de)

A wide range of studies (e.g., refs. [1], [2]) have shown that most of tropical Africa north of about 10° S was drier during the Last Glacial Maximum (LGM; 23–19 ka), relative to today, and wetter during the early to mid Holocene, which has been defined[3], [4] as the African Humid Period (AHP; ca. 11.5–5.5 ka). Abrupt precipitation changes during the glacial and deglacial are associated with major changes in the Atlantic Meridional Overturning Circulation (AMOC) and sea surface temperature patterns[2], [5]. However, a large and abrupt aridification, with respect to gradual precessional insolation forcing, has also been documented at some sites during the Holocene at ~5.5 ka (the AHP termination)[3], [4], the causes of which are currently unresolved. An abrupt AHP termination was originally thought to have been caused by a collapse of Saharan and Sahelian vegetation at 5.5 ka[6] switching the climate to an arid equilibrium state. Many vegetation records, however, do not show a collapse[4], [7] and the latest coupled climate models[8] suggest the positive biogeophysical feedback was not strong enough to have triggered an abrupt climate switch. Complicating the picture, many hydrological records suggest a gradual (e.g., ref. [7]) or time-transgressive[9], [10] aridification at the AHP termination, more in line with a direct and linear response to precessional insolation forcing. Moreover, some intermediate complexity model simulations (e.g., ref. [11]) have difficulty in simulating an abrupt AHP response and most fully coupled models underestimate the intensity of precipitation during the AHP[9], [12]. Overall, it is not resolved whether a rapid termination of the AHP was ubiquitous and synchronous at 5.5 ka, why this took place at 5.5 ka, and whether additional feedbacks or teleconnections were involved.

Precipitation in tropical Africa results from a combination of factors including the monsoonal on-land flow of moist air, low-level convergence of air at the intertropical convergence zone and, of particular importance, the deep vertical motion of air, which over northern Africa is modulated by the interaction of the Tropical Easterly Jet (TEJ) and the African Easterly Jet (AEJ)[13]. These jets oscillate seasonally and at present reach maximum latitudes of 6–8° N (TEJ) and 14–17° N (AEJ) in August[13]. The TEJ maximum wind-speed is in the upper troposphere at ~150 hPa, while the AEJ maximum windspeed is in the mid-troposphere at ~600 hPa. The TEJ extends from India across the African continent (Fig. 1a, Supplementary Fig. 1) and is maintained by the upper tropospheric temperature gradient between the equatorial latitudes and the relatively

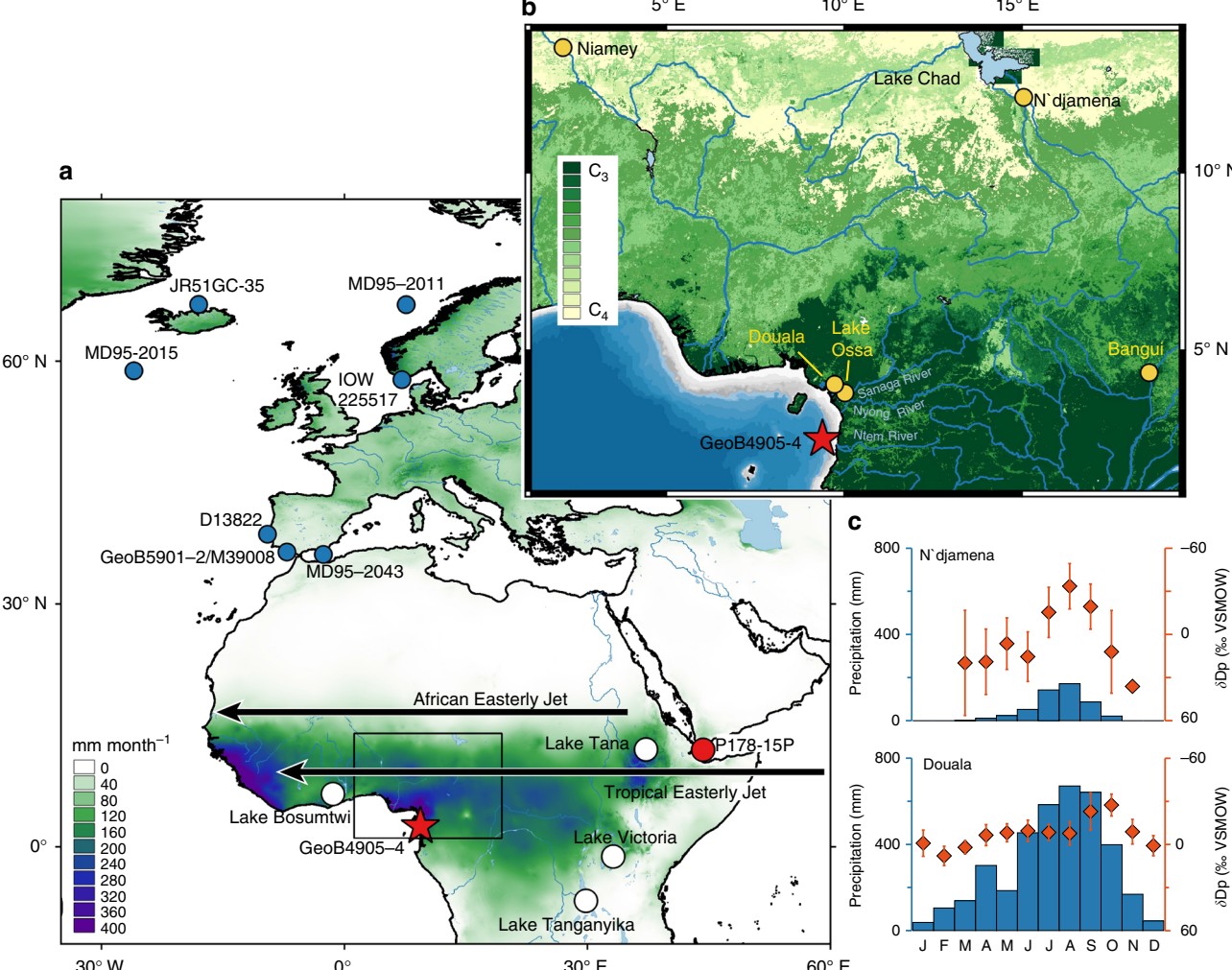

**Fig. 1** Maps of the study area and climatology. **a** Colours represent mean monthly precipitation (mm) for the months Jun to Oct, the primary wet seasons for southern Cameroon and the Sahel. Red star marks the study site GeoB4905-4 (2°30.0´ N, 09°23.4´ E) in the Gulf of Guinea. Red dot marks the Gulf of Aden P178-15P core site[4], white dots mark other sites discussed in the text and blue dots mark SST records from Supplementary Table 1. Black arrows mark position of TEJ and AEJ in summer[13]. Black box marks the inset. **b** Zoomed-in map of the study region showing C₃–C₄ vegetation distribution, rivers and bathymetry. Yellow dots mark the Douala, N'djamena, Niamey and Bangui GNIP stations, and Lake Ossa. Bathymetry shallower than 120 m is coloured in grey. **c** Monthly precipitation amount and $\delta D_p$ data for N'djamena, Chad and Doula, Cameroon[27], highlighting the large seasonal $\delta D_p$ changes in the Sahel compared to equatorial regions. Error bars represent standard deviation (1σ) of monthly measurements

warmer subtropics[14]. A slower TEJ is associated with drier conditions in these regions, due to reduced upper-level divergence and hence reduced upward vertical flow[13, 15, 16]. The AEJ is attributed to the meridional temperature gradient in the Sahel and a faster AEJ results in greater moisture export and drier conditions in the western Sahel[13, 17]. The African rainbelt oscillates across southern Cameroon twice a year, bringing most precipitation during northern hemisphere autumn (Sep–Oct–Nov; SON) and some during spring (Mar–Apr–May; MAM), while northern Cameroon and the Sahel receive precipitation primarily during summer (Jun–Jul–Aug; JJA; Fig. 1a).

Sedimentary leaf-wax $n$-alkane $\delta D$ ($\delta D_{wax}$) has been shown to primarily reflect precipitation $\delta D$ ($\delta D_p$) in Cameroon and globally, and in the tropics is often taken to reflect precipitation amount[18]. While biosynthesis of leaf-wax $n$-alkanes is thought to exert a constant hydrogen isotope fractionation against leaf water, secondary controls on $\delta D_{wax}$ include relative humidity and vegetation type[13]. $\delta D_{wax}$ from $C_4$ grasses is less sensitive to transpirational D enrichment in plant leaves, likely due to partial use of unenriched xylem water in $n$-alkane synthesis[19]. Other plant physiological differences such as the water source available to the plant and seasonal timing of leaf-wax biosynthesis may also influence $\delta D_{wax}$ values[18]. Higher relative humidity is thought to reduce evapotranspirational isotopic enrichment of leaf and soil waters, so that in the tropics relative humidity variability tends to amplify the $\delta D_{wax}$ variability that is driven by the amount effect[18].

Sedimentary $\delta^{13}C_{wax}$ is often used as an indicator of $C_3$ and $C_4$ vegetation-type changes. African $C_3$ trees, shrubs, herbs and lianas

$(n = 45)$ exhibit a mean ($C_{29}$ $n$-alkane) $\delta^{13}C_{wax}$ value of $-35.7\permil \pm 2.9\permil$[20] while African $C_4$ grasses ($n = 38$) exhibit a mean $\delta^{13}C_{wax}$ value of $-21.4\permil \pm 2.0\permil$[21]. Much of the catchments of the Ntem, Nyong and Sanaga Rivers are dominated by $C_3$ trees (Fig. 1b)[22] and this is reflected in surface sediments of Lake Ossa, southern Cameroon (Fig. 1b), which exhibit a $\delta^{13}C_{wax}$ value of $-35.4\permil$[23]. Conversely, further north, the Sahel-Sahara and much of the Niger River catchment are dominated by $C_4$ plants (Fig. 1b)[22], and this is evident in marine sediments off West Africa[24].

To provide more insights into the AHP termination, we assess large-scale hydroclimatic changes in Cameroon and the central Sahel-Sahara using $\delta D_{wax}$ from a marine sediment core GeoB4905-4 in the Gulf of Guinea (Figs. 1a, b). We also assessed $\delta^{13}C_{wax}$ as an indicator for $C_3$ vs. $C_4$ vegetation type. Our results indicate high precipitation during the AHP followed by a rapid precipitation decrease at 5.8-4.8 ka, similar to a record from northern East Africa[4]. We show that the rapid precipitation decrease was likely triggered by northern high-latitude cooling. The cooling reduced the speed of the TEJ, triggering rainfall reduction that was amplified by climate feedbacks and resulted in strong aridification over a relatively short period.

## Results

**Moisture sources.** To assess the likely moisture sources to present-day southern Cameroon, we performed analyses using the 3-D Lagrangian model FLEXPART[25]. The backward airmass trajectories (Fig. 2a–d) indicate the southeast Atlantic and central

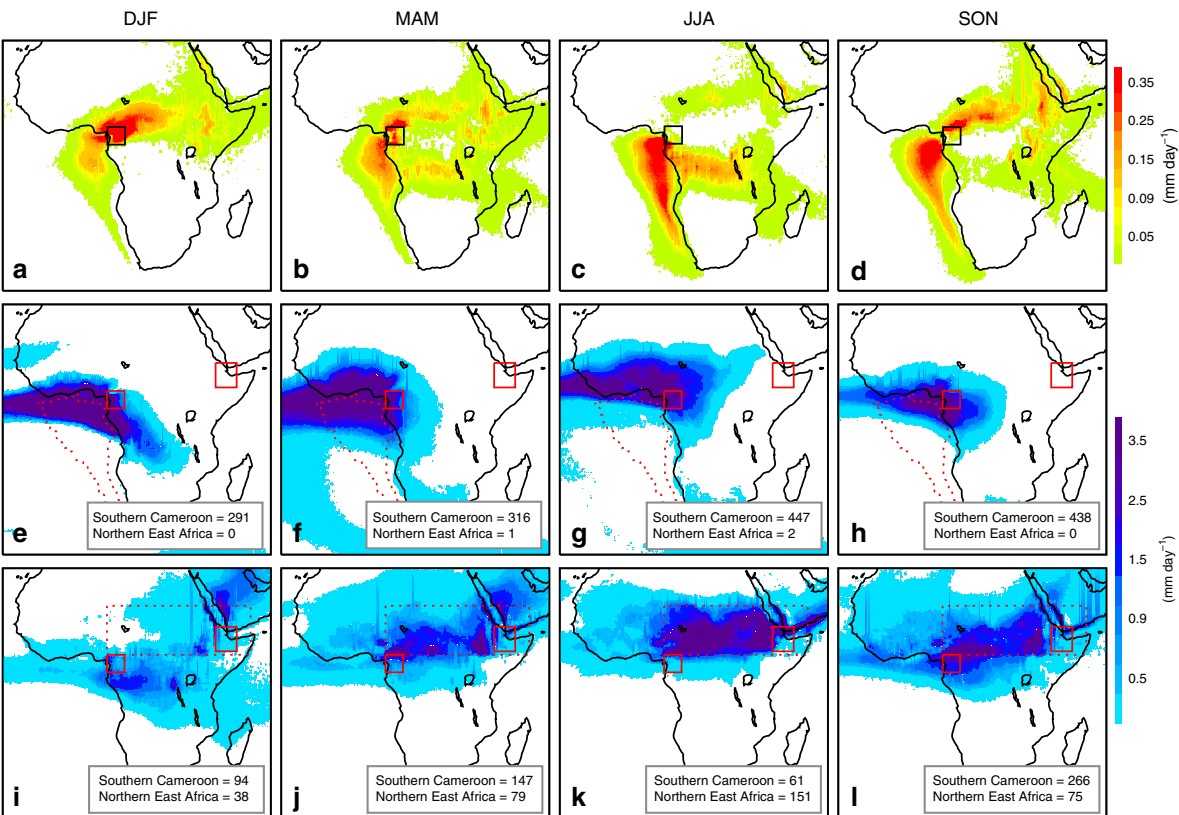

**Fig. 2** Moisture sources for southern Cameroon and northern East Africa. **a–d** FLEXPART[84, 85] backward analyses of air mass trajectory for the period 1980–2015 at 0.25° resolution. The boxed region in southern Cameroon (9° E-14° E and 1° N-6° N) represents the estimated leaf-wax source region for Gulf of Guinea core GeoB4905-4. Colours represent the sources of moisture for the boxed region and show where E-P > 0 (mm day⁻¹). **e–h** Forward runs of FLEXPART for the southeast Atlantic moisture source (outlined with a red dashed line). Colours show precipitation derived from this moisture source (mm day⁻¹). Numbers indicate total seasonal precipitation amount (mm) from this moisture-source delivered to the boxed regions in southern Cameroon and northern East Africa. The boxed region in northern East Africa represents the leaf-wax source region for the Gulf of Aden core P178-15P, estimated as 40° E-46° E and 7° N-14° N. **i–l** As **e–h** but for the Sahel-Sahara moisture source (9° E-50° E and 6° N-20° N; marked with a red dashed rectangle). **a**, **e**, **i** represent Dec–Jan–Feb (DJF), **b**, **f**, **j** represent Mar–Apr–May (MAM), **c**, **g**, **k** represent Jun–Jul–Aug (JJA) and **d**, **h**, **l** represent Sep–Oct–Nov (SON)

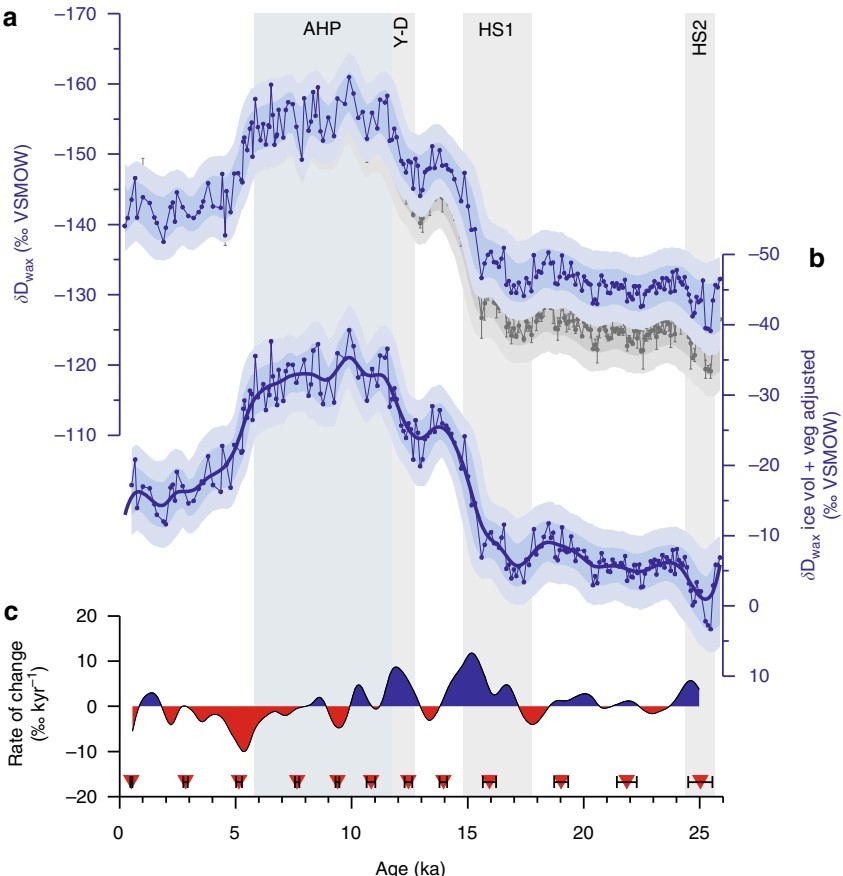

**Fig. 3** $\delta D_{wax}$ from core GeoB4905-4 in the Gulf of Guinea. **a** Grey timeseries represents unadjusted $\delta D_{wax}$. Error bars are individual analytical uncertainty. Shadings indicate 68 and 95% uncertainty bounds, including a mean analytical uncertainty of 3‰ and age uncertainty. Blue timeseries represents $\delta D_{wax}$ adjusted for ice volume, shading as above. **b** $\delta D_{wax}$ adjusted for ice-volume and vegetation-type changes, representing an estimate of past $\delta D_p$. Shadings as above. Thick black line is the Ruppert-Sheather-Wand smooth, the optimal smoothing for the data set[86]. **c** Rate of change (‰ kyr$^{-1}$) based on Ruppert-Sheather-Wand smooth. Blue colours representing periods of wettening, red represent periods of aridification. Red diamonds mark calibrated radiocarbon age control points. Vertical bars highlight the African Humid Period (AHP), Younger-Dryas (Y-D) and Heinrich Stadials 1 and 2 (HS1, HS2)

Sahel-Sahara to be the major moisture sources to southern Cameroon during the SON and MAM seasons. Forward analyses for the southeast Atlantic (Fig. 2e–h) and Sahel-Sahara (Fig. 2i–l) moisture sources reveal the spatial distribution and amount of precipitation that is generated by the moisture derived from these two sources. This shows that the southeast Atlantic and Sahel-Sahara moisture sources contribute 438 mm and 266 mm of precipitation to southern Cameroon, respectively, for SON season, and 1492 mm and 568 mm over the year.

**$\delta D_{wax}$ and $\delta^{13}C_{wax}$ variability**. We focus on the $C_{29}$ $n$-alkane, denoted as $\delta D_{wax}$ and $\delta^{13}C_{wax}$ (Supplementary Notes 1 and 2). The $\delta^{13}C_{wax}$ values from GeoB4905-4 are generally low and display small variability, ranging between −33.5‰ and −30.3‰ (Supplementary Fig. 2). $\delta D_{wax}$ values have been adjusted for the effect of ice-volume and vegetation-type changes (Methods section; Fig. 3a, b), although this has a minor effect on the climate signal. The adjusted $\delta D_{wax}$ record (Fig. 3b) displays large variability and three main transitions. $\delta D_{wax}$ values were higher during the LGM (23–19 ka) relative to today. As indicated by SiZer analysis (Methods section; Fig. 4), this is followed by two periods of significant $\delta D_{wax}$ decrease: between 15.9 and 13.9 ka (the end of Heinrich Stadial 1; HS1) and between 12.5 and 11.5 ka (the end of the Younger-Dryas; Y-D). The mean rates of change for these two transition periods are 8‰ kyr$^{-1}$ and 7‰

kyr$^{-1}$, respectively (Fig. 3c). $\delta D_{wax}$ values remained low between 11.5 and 5.8 ka, corresponding to the AHP. Between 5.8 ka and 4.8 ka the record shows a significant $\delta D_{wax}$ increase (Figs. 3b, c and 4), associated with the AHP termination. In particular, there is a significant increase at lower bandwidths at 5.3 ka, indicating a particularly rapid drying at this time (Fig. 4). The mean rate of change during the AHP termination (5.8 ka and 4.8 ka) is 8‰ kyr$^{-1}$.

**Origin of the $\delta D_{wax}$ signal**. Relatively low $\delta^{13}C_{wax}$ values (mean of −32.3‰, range from −33.5‰ to −30.3‰; Supplementary Fig. 2) over the past 25 kyr suggest that leaf waxes were mainly derived from $C_3$ vegetation. This agrees with previous work[26] that the catchments of the Ntem, Nyong and Sanaga Rivers were the main source region of leaf-wax $n$-alkanes to the core site (Supplementary Note 2). Nonetheless, our data indicates a slightly higher $C_4$ contribution than Lake Ossa surface sediment (−35.4‰; ref.[23]) especially during the late Holocene. This points to an additional $C_4$ contribution to the marine sediment that was probably delivered to the core site as Sahelian-Saharan dust (from, for example, the Bodélé depression) and/or Niger River material. Based on linear mixing with the above $C_3$ and $C_4$ end-members, $\delta^{13}C_{wax}$ values over the last 25 kyr would correspond to mean a $C_4$ vegetation contribution of 24%, with a range between 15 and 41%.

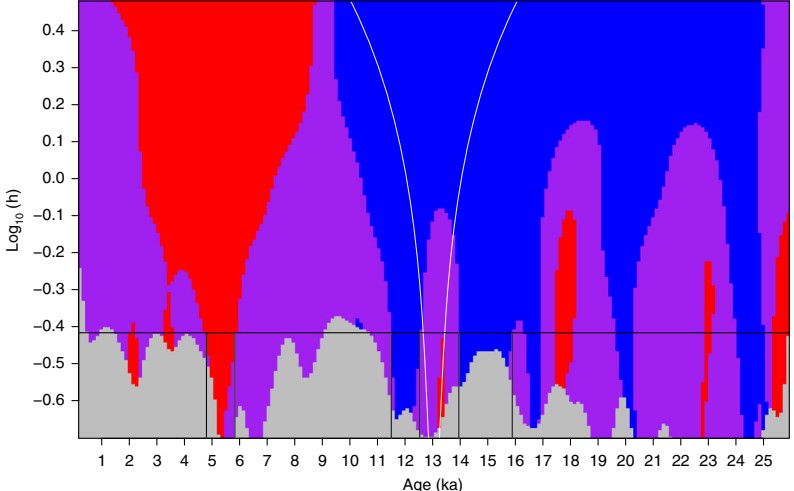

**Fig. 4** SiZer map for GeoB4905-4 $\delta D_{wax}$. The *y*-axis represents the range of bandwidths (*h*) for which the data were smoothed (plotted on a log scale) and the *x*-axis represents age. Blue regions represent significant decreases in $\delta D_{wax}$ (wettening), red regions significant increases in $\delta D_{wax}$ (drying), purple regions no significant change, and grey areas indicate where the sampling resolution is too low. The black horizontal line represents the data-driven Ruppert-Sheather-Wand bandwidth[86], the optimal smoothing (global bandwidth) for the entire data set. Time intervals where this line intersects with areas of significant increase or decrease are highlighted with vertical lines. $\delta D_{wax}$ is adjusted for ice volume and vegetation type

The minor $\delta^{13}C_{wax}$ variability (Supplementary Fig. 2) suggests that vegetation type is unlikely to be the main control on $\delta D_{wax}$, particularly for the large magnitude change between 5.8 and 4.8 ka, when $\delta^{13}C_{wax}$ shows little change. Rather than changes in vegetation, the $\delta D_{wax}$ record reflects changes in $\delta D_p$. Tropical $\delta D_{wax}$ records are commonly interpreted as being controlled by the amount effect (e.g. ref. [4]) with higher $\delta D_{wax}$ values representing drier conditions. Given that most leaf waxes originate from southern Cameroon with a smaller contribution from the Sahel, they likely reflect mainly southern Cameroon $\delta D_p$ and partly Sahel $\delta D_p$. However, the amount effect can operate locally and non-locally, i.e., $\delta D_p$ from southern Cameroon can reflect the amount effect 'upstream' in the moisture-source region (thus integrating over a larger area than that of the leaf-wax source region). Given that annually almost 30% of the moisture in southern Cameroon originates from the central Sahel-Sahara (Fig. 2), it suggests $\delta D_p$ in southern Cameroon is significantly affected by hydroclimatic processes in the Sahel-Sahara. A further consideration is that the relationship between precipitation amount and $\delta D_p$[27] is steeper at sites in the semi-arid regions of the Sahel compared to the equatorial regions (Supplementary Fig. 3), implying past precipitation changes in the Sahel would cause a larger $\delta D_p$ change than precipitation changes in southern Cameroon, potentially over-printing the southern Cameroon signal. Thus, changes in $\delta D_{wax}$ likely reflect integrated changes in precipitation amount in both southern Cameroon and the central Sahel-Sahara.

To understand the upstream signal over time, we investigated $\delta D_p$ over the last 25 ka using a transient simulation of the intermediate complexity isotope-enabled climate model *iLOVE-CLIM* (Methods section). The transient simulation displays a similar evolution of atmospheric $\delta D_p$ in southern Cameroon and the Sahel-Sahara (Supplementary Note 3), but a different evolution of precipitation amount in the two regions (Supplementary Fig. 4b–e). This suggests that in this model, southern Cameroon $\delta D_p$ is reflecting an integrated precipitation amount signal from both Cameroon and the central Sahel-Sahara.

**Rapid deglacial and Holocene hydrological changes.** The Gulf of Guinea $\delta D_{wax}$ record suggests slightly drier conditions at the LGM compared to the late Holocene (Fig. 3b). Cooler conditions

at the LGM would suggest that the magnitude of aridification at the LGM relative to the late Holocene is likely to be conservative (Methods section). Drier LGM conditions are in line with most other hydroclimate records from northern Africa (e.g., ref. [1]). Increased precipitation at the terminations of HS1 and the Y-D (Fig. 3b) is seen in other records across much of northern Africa north of ~10° S[2]. Both rapid increases are attributed to: $CO_2$-driven deglacial tropical SST increase and atmospheric warming, increasing the moisture content of the atmosphere and; to AMOC resumption and northern high-latitude SST increase, allowing the rainbelt to penetrate further northwards[2].

A more surprising finding in our Gulf of Guinea $\delta D_{wax}$ record is the rapid aridification between 5.8 and 4.8 ka with a particularly sharp drop at 5.3 ka (Figs. 3b, c and 4), which exhibits comparable rate of change and duration to changes at the termination of HS1 and the Y-D. Although our $\delta D_{wax}$ implies a large aridification between 5.8 and 4.8 ka, salinity changes in the Gulf of Guinea, which reflect Ntem, Nyong and Sanaga River discharge in southern Cameroon, display a smaller increase around this time[28]. This would suggest rapid aridification at the AHP termination was more prominent in the Sahel-Sahara than in southern Cameroon.

**$\delta D_{wax}$ records from other regions.** The rapid aridification at the AHP termination is similar to that observed between 5.4 ka and 4.5 ka[4] in the Gulf of Aden, northern East Africa (Fig. 5b, c). The transitions are coeval, within the age uncertainty of the records (±0.3 kyr between 5.8 and 4.8 ka for the Gulf of Guinea record; ±0.1 kyr between 5.4 ka and 4.5 ka for the Gulf of Aden). The mean rate of change for the transition period is 5‰ kyr$^{-1}$ for the Gulf of Aden record, comparable to the Gulf of Guinea record. The leaf-wax source region for the Gulf of Aden record mainly receives rainfall during JJA, and also receives a significant contribution of Sahel-Sahara moisture (Fig. 2i–l), suggesting that the rapid AHP termination was a feature spanning the latitudes of the Sahel during the JJA season. Given wetter conditions in the Sahel and Sahara during the mid-Holocene (e.g., refs. [1, 29]), it is plausible that the Sahel-Sahara was a more important moisture source for Cameroon and northern East Africa during the AHP compared to today.

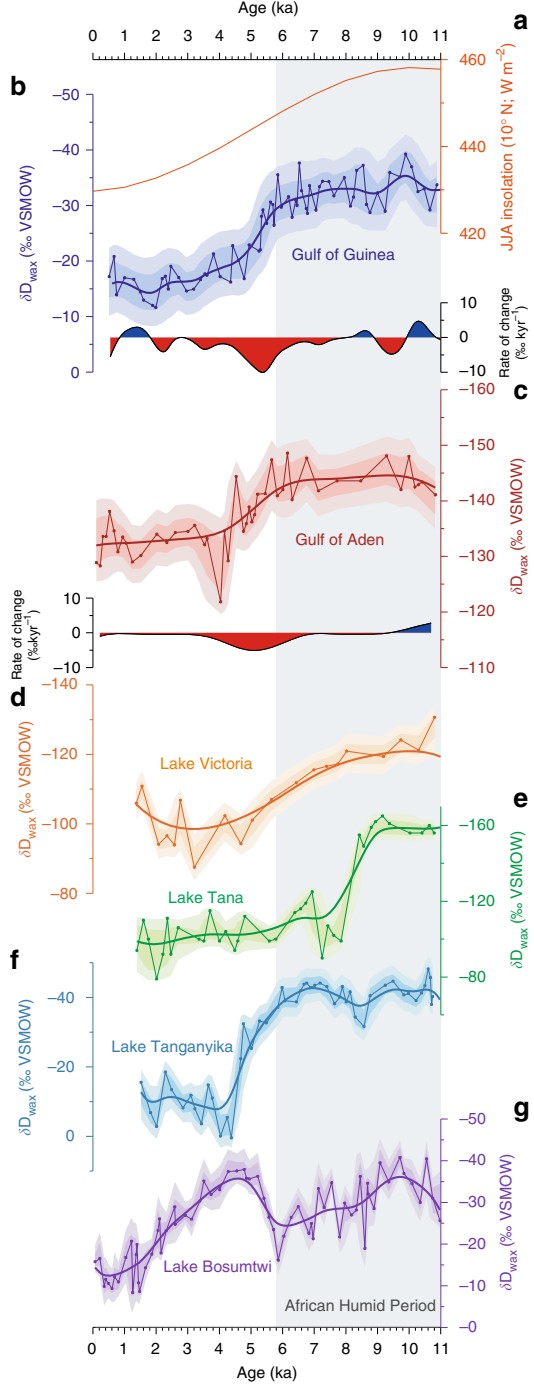

**Fig. 5** Comparison with other African $\delta D_{wax}$ records. **a** Mean JJA insolation at 10° N. **b** Gulf of Guinea $\delta D_{wax}$ from core GeoB4905-4 (based on the $C_{29}$ n-alkane; ice-volume and vegetation adjusted; this study). **c** Gulf of Aden $\delta D_{wax}$ (based on the $C_{30}$ fatty acid; ice-volume adjusted) from core P178-15P[4]. **d** Lake Victoria $\delta D_{wax}$ (based on the $C_{28}$ fatty acid; ice volume adjusted)[30]. **e** Lake Tana $\delta D_{wax}$ (based on the $C_{28}$ fatty acid; ice volume adjusted)[31]. **f** Lake Tanganyika $\delta D_{wax}$ (based on the $C_{28}$ fatty acid; ice volume and vegetation adjusted)[33]. **g** Lake Bosumtwi $\delta D_{wax}$ (based on the $C_{31}$ n-alkane; ice volume and vegetation adjusted)[10]. Shadings indicate 68 and 95% uncertainty bounds, including analytical and age uncertainty. Thick lines represent Ruppert-Sheather-Wand smooth: rate of change (‰ kyr$^{-1}$) in **b** and **c** is based on this smooth

Other $\delta D_{wax}$ records from East Africa sometimes show a different evolution at the AHP termination, likely attributable to the seasonality of precipitation and/or moisture-source variability. Lake Victoria displays a relatively gradual $\delta D_{wax}$ increase from the early to late Holocene (Fig. 5d)[30]: the absence of a rapid change at around 5.5 ka may be because the main wet season at this site is during MAM, and thus $\delta D_p$ is unlikely to be influenced by Sahel-Sahara JJA moisture. Lake Tana (Fig. 5e) displays a rapid and large magnitude increase at ~8.5 ka, attributed to a reduction in Congo-basin derived recycled moisture[31]. This record displays, however, little $\delta D_{wax}$ change at 5.5 ka, although sedimentary Ti does show a major decrease at 5.5 ka[32], indicating aridification and perhaps highlighting complex moisture-source effects on $\delta D_{wax}$ at this site. Lake Tanganyika (Fig. 5f) in eastern Central Africa displays a large and rapid $\delta D_{wax}$ increase between 5.7 ka and 4.4 ka[33], in line with our record. Lake Tanganyika is located well south of the Sahel, and receives a minor amount of Sahel-Sahara moisture (Fig. 2i-l). The rapid $\delta D_{wax}$ increase may reflect local aridification, or, given that Lake Tanganyika is also susceptible to E-W moisture shifts[34], may reflect central African moisture-source changes.

In West Africa, the crater lake Bosumtwi $\delta D_{wax}$ record[10], was interpreted as reflecting reduced precipitation between ~9 ka and 5.5 ka, followed by a return to wet conditions at 5.5 ka and then termination of the AHP at ~3.5 ka (Fig. 5g). Lake Bosumtwi $\delta D_{wax}$ disagrees, however, with the Bosumtwi lake-level record, which was 110 m higher than today and overflowing the crater rim between 9 ka and 5.7 ka, followed by a lake-level decrease at some point between 5.7 ka and ca. 2.0 ka[10]. The lake-level decrease is thought to have resulted in input of material from the crater walls, as observed in radiocarbon measurements of the late Holocene[35]. Thus, it seems possible that post-highstand Bosumtwi $\delta D_{wax}$ may be partly biased by input of pre-aged leaf waxes, which could explain the difference to the GeoB4905-4 $\delta D_{wax}$ record. Offshore NW Africa, $\delta D_{wax}$ records have shown wet conditions during the AHP; in particular core GC37 displays aridification at about 5.5 ka[9], similar to our record. This was interpreted as a rapid response at the AHP termination, although bioturbation was thought to be significant in these lower-resolution records, making direct comparisons difficult.

**Insights from other hydrological proxies.** Other proxies from Africa also sometimes show spatially variable responses at the AHP termination. The vegetation record of Lake Yoa was interpreted as representing a gradual aridification through the Holocene[7]. Persistence of wet conditions in this region after the AHP has, however, been attributed to the Tibesti Mountains acting as a 'water tower'[36]. Compilations of past hydrology spanning the Sahel-Sahara have been interpreted as showing a heterogeneous response, with north-south[10] and east-west[29] differences in the timing of aridification. The compilations are, however, partly based on discontinuous records, that are less well dated compared to marine records, and include a range of different hydrological indicators, which together might explain this heterogeneity. Nonetheless, from our record we cannot rule out that the northernmost Sahara[10] dried earlier than the southern Sahara and Sahel.

In support of our record, several other proxies provide additional evidence for a rapid end to the wet conditions of the AHP in the Sahel-Sahara. The lake-level record of Lake Mega-Chad displays high levels until ~5.2 ka, when the water balance rapidly decreased[36]. In NW Africa, the ODP658C dust record shows a very abrupt increase at 5.5 ka[3], and other dust flux records show increases at around 4.9 ka[37], although we note that dust may not necessarily be directly related to hydrology. In

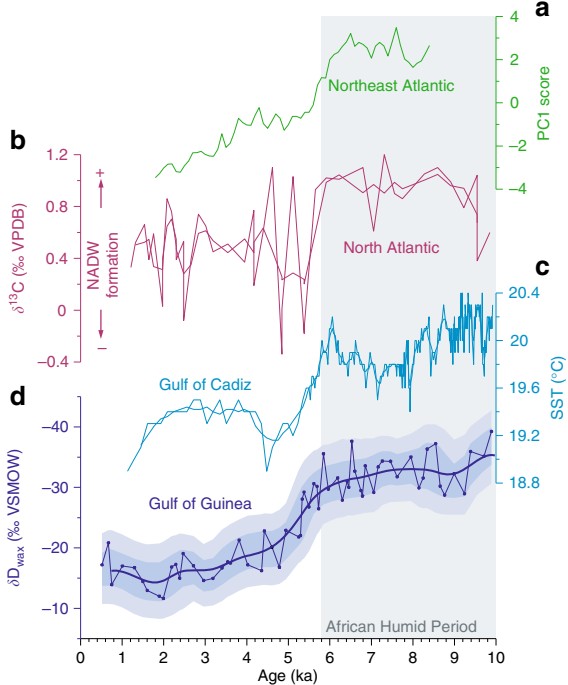

**Fig. 6** Comparison with mid- and high-latitude records. **a** Principle Component 1 (PC1) scores from eight alkenone SST records in the northern Atlantic (Supplementary Table 1). PC1 represents 57.8% of the variance. **b** $\delta^{13}C$ of benthic foraminfera from core EN120-GGC1 in the north Atlantic, interpreted as a record of NADW formation and ocean circulation: lower values represent slower ocean circulation[54]. **c** Alkenone SST from core GeoB5901-2 in the Gulf of Cadiz, just to the north of Africa[87]. **d** $\delta D_{wax}$ from GeoB4905-4 (ice-volume and vegetation adjusted, this study). Shadings indicate 68 and 95% uncertainty bounds, including analytical and age uncertainty

northern East Africa a large increase in the deposition of K-rich sediment, is evident between ~5.8 and 4.8 ka at Chew Bahir[38], indicating aridification, similar to the drop in Ti at 5.5 ka at Lake Tana[32]. Also in northern East Africa, lake levels at Lakes Abhe[39], Zibay Shalla[40] and Abiyata[41] display major decreases at about 4.5 ka, 5.0 ka and 5.4 ka, respectively. Therefore, overall, a number of records lend support to the hypothesis of a rapid AHP termination at about 5.5 ka covering the Sahel-Sahara and northern East Africa.

**The role of biogeophysical feedbacks.** Hydroclimate stability in the Sahel-Sahara during the AHP followed by a rapid aridification at 5.8–4.8 ka would not be in line with a response to local precessional insolation forcing, which began to decrease at around 9 ka (Fig. 5a). This raises the question of why climate remained wet until 5.8 ka and what caused the large-scale response at this time. Either internal climate feedbacks created a non-linear response to the external forcing due to a threshold in the system, or there was a teleconnection driving the rapid aridification beginning at 5.8 ka.

Sahel-Sahara vegetation and soil moisture are thought to exert a positive feedback effect, i.e., enhancing wetter conditions during the AHP[8]. However, these feedbacks are considered too weak to have caused a tipping point[8], and thus would not themselves have been the initial trigger for the onset of aridification at 5.8 ka. Nonetheless, we do not rule out that these positive feedbacks enhanced the rate of aridification at the AHP termination once underway. Atmospheric dust is also believed to have been an important feedback in enhancing the wetness during the AHP[42],

and thus was potentially another factor enhancing aridification at the AHP termination.

Models indicate that lakes and wetlands may also constitute a positive feedback[43] via modulation of the regional moisture balance. The rapid 100 m depth decrease of Lake Mega-Chad at about 5.2 ka[36] would have reduced the lake area from the maximum estimated extent of 350,000 km[244] towards the 'pre-industrial' [1960] value of 25,000 km[2], perhaps reducing moisture contribution and enhancing the rapidity of AHP termination. Nonetheless, other studies suggest that the positive feedback from Lake Chad was weak due to the cool lake surface inhibiting deep convective precipitation[45] and thus it also seems unlikely that lakes and wetlands were the sole trigger for the AHP termination.

An additional potential mechanism invokes tropical SST[4]. It was suggested that Indian Ocean SST decreased below a critical threshold at ~5.0 ka, substantially reducing tropical East African precipitation. However, SST records from both the western Indian Ocean (Supplementary Fig. 5a–c) and Gulf of Guinea (Supplementary Fig. 5d) do not show a significant SST change at this time, suggesting that tropical SSTs were not the trigger for the rapid precipitation decrease on the eastern or western sides of the continent.

Overall, models suggest that vegetation, soil moisture, dust, lake and wetland feedbacks, were not the critical trigger tipping the climate towards a drying state. It is possible that the models are simply deficient in representing these processes. Alternatively, it is possible that a trigger was needed from further afield within the climate system, to initiate the onset of feedbacks and the AHP termination. Because we see rapid aridification on the east and west sides of Africa (Fig. 5b,c), this trigger was likely teleconnected to a large-scale atmospheric circulation feature, such as the TEJ. We suggest that a TEJ slowdown was triggered by a cooling of the Northern Hemisphere mid- and high-latitudes.

**High-latitude cooling triggered the AHP termination.** A range of records from the northern high latitudes including Greenland[46], the Norwegian sea[47, 48] and the Fram Strait[49, 50] indicate a rapid drop in summer temperature between ~6.0 and 5.0 ka. Other records also indicate an increase in Arctic sea ice[50, 51] at about 5.5 ka. Empirical Orthogonal Function (EOF) analysis of temperature records from Canada and Greenland suggests an onset of rapid cooling at ~5.0 ka[52]. We performed EOF analysis of Holocene alkenone SST records from the Arctic and northeast Atlantic (Supplementary Table 1, Fig. 1a). Although the overall trend of the Principle Component 1 from the analysis is one of gradual insolation driven cooling, it does indicate more rapid cooling between about 6.0 and 5.5 ka than earlier or later in the Holocene (Fig. 6a). Rapid north Atlantic cooling at this time may have been related to an AMOC slowdown between ~6.0 and 5.0 ka[53–55] (Fig. 6b). High-latitude cooling may also have been linked to increased Arctic sea-ice generation, that has been attributed to sea-level induced flooding of the Laptev Sea shelf at ~5.0 ka[49, 56]. Other studies suggested that between 6.0 and 5.0 ka an expanded polar vortex[57] brought winter-like conditions to the mid-latitudes, evident in north America[58]. Cooling is evident in Europe[59] and SSTs just to the north of Africa (Fig. 6c), suggesting that the cool anomaly expanded from the mid-high latitudes towards northern Africa with eastern boundary currents.

To investigate how such a northern extratropical cooling affected African hydrology, we used a high-resolution version of the fully coupled CCSM3 climate model (Methods section). We simulated AHP conditions with an early Holocene (EH, 8.5 ka) control run, and subsequently initiated an extratropical North Atlantic cooling by a freshwater-induced slowdown of the AMOC (experiment EH$_{fre}$). Note that a freshwater perturbation is a

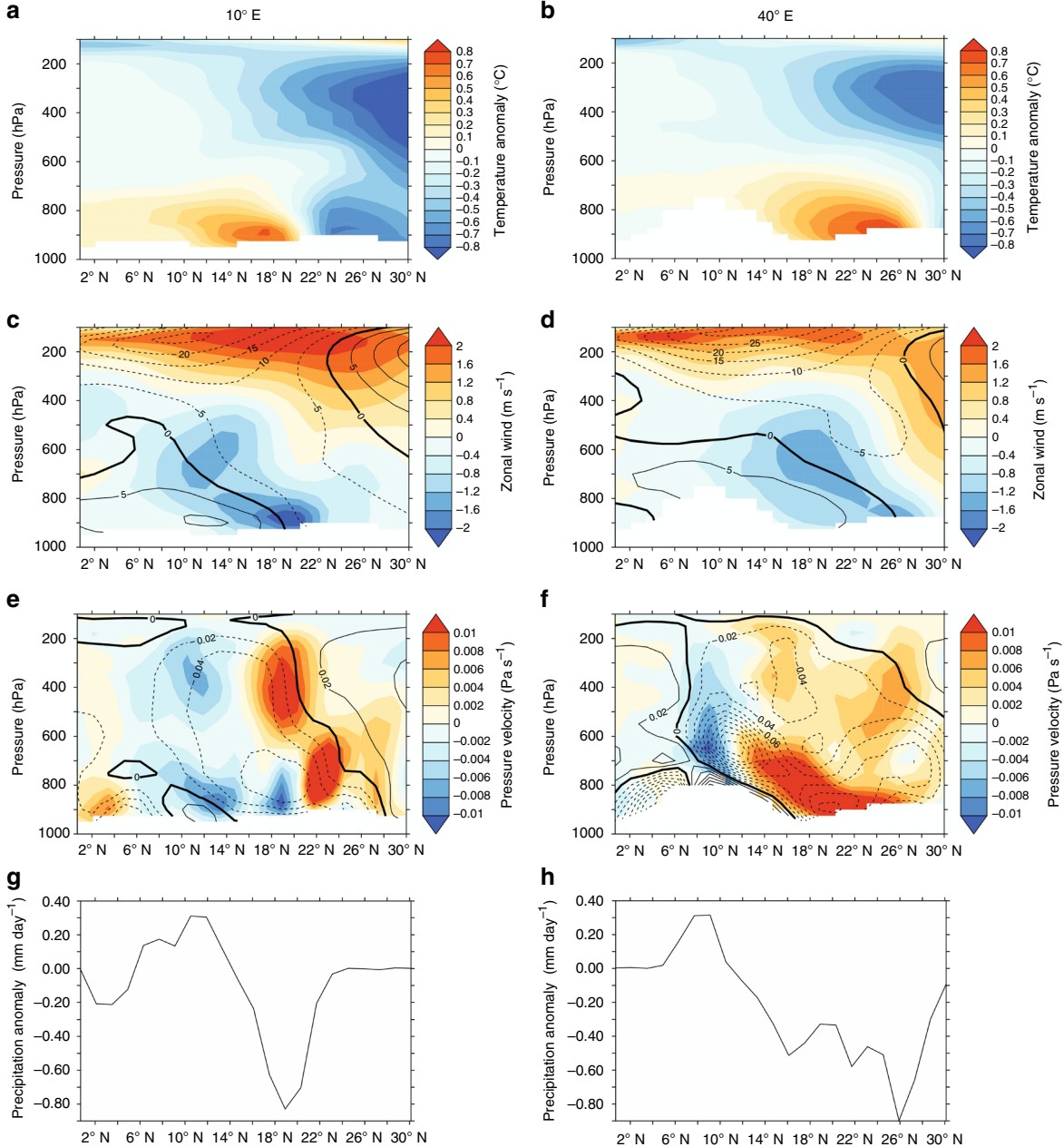

**Fig. 7** CCSM3 model output showing the effect of north Atlantic cooling on northern African winds and precipitation during the AHP. Height-latitude plots along longitudes 10° E (**a**, **c**, **e**, **g**) and 40° E (**b**, **d**, **f**, **h**) during JJA season. **a**, **b** Tropospheric temperature anomalies (°C) for the EH$_{fre}$–EH experiment, highlighting the cool anomaly over the northern Sahara. **c**, **d** Zonal wind speed (m s$^{-1}$), with contours representing the early Holocene control run (EH; negative values represent easterly winds). Shading represents anomalies (EH$_{fre}$—EH): red region around 150 hPa is a negative easterly anomaly, blue region a positive easterly anomaly. **e**, **f** Vertical flow (Pa s$^{-1}$), with contours representing the EH control run (negative values represent upward motion). Shading represents anomalies (EH$_{fre}$—EH): red shows decreased upward motion, blue increased upward motion. **g** Precipitation anomalies (EH$_{fre}$–EH; mm day$^{-1}$) along 10° E. **h** Precipitation anomalies (EH$_{fre}$-EH; mm day$^{-1}$) along 40° E

simple and common method to induce a cooling in the high northern latitudes and we do not imply a large input of freshwater at this time. In experiment EH$_{fre}$, surface temperatures decrease by 0.5–2.5 °C in the northeastern North Atlantic compared to EH. The EH simulation clearly shows the TEJ at ~10° N and 150 hPa and the AEJ at ~20° N and 500–600 hPa (contours in Fig. 7c, d). The simulated EH$_{fre}$–EH anomaly shows that the high- and mid-latitude JJA cooling (Fig. 7a, b) reaches northern Africa. The cool anomaly is evident throughout the troposphere in the northern Sahara from the western-to-central (10° E) and eastern (40° E) regions (Fig. 7a, b), acting to reduce the meridional gradient of upper tropospheric temperature

between the Sahara and the equatorial latitudes. In accordance with the thermal wind relation, this weakens the TEJ (red shading in Fig. 7c, d), leading to reduced upper-level divergence. In the western-to-central region (10° E; Fig. 7c) the slowdown of the TEJ is particularly pronounced at its anticyclonic poleward flank, where the upper-level divergence is usually strongest[60]. This reduces upward vertical motions in the mid to upper troposphere at 16–23° N at 10° E and north of 12° N at 40° E (red and orange regions in Fig. 7e, f) driving a reduction in precipitation at similar latitudes of the Sahel-Sahara (Fig. 7g, h). In addition to the upper tropospheric dynamical processes, surface cooling in the Saharan region is associated with a weakening of the Sahara Heat Low[61],

which reduces the westerly inflow and northward penetration of low-level moist monsoon winds (Fig. 7c, d) and hence moisture convergence. Drier conditions are further associated with a reduced low-level moist static energy and hence a more stable atmosphere, hampering deep convection[61, 62]. These mechanisms strongly agree with previous modern-day model experiments and instrumental/re-analysis data[61–63], although in these experiments, the main area of drying was located further south in the Sahel and central Africa[61], as would be expected in a situation when the Sahara is arid. In the model data[63] it was found that the changes in the TEJ and Sahara Low tend to precede the change in precipitation, suggesting them to be a cause of rather than response to the change in Sahel rainfall. Additionally, during aridification, a shift of the soil moisture and meridional surface temperature gradient has been shown to strengthen the AEJ, further reducing rainfall across the west and central Sahel[17, 64]: thus the AEJ may represent an additional feedback contributing to the rapid aridification at 5.8–4.8 ka. Furthermore, models suggest that the high-latitude cooling was enhanced by the African precipitation decrease[65], and this connection may have contributed to a tipping point behaviour of the two regions.

## Discussion

In comparison with the high- and mid-latitude temperature decrease at the AHP termination, the increase at the AHP onset was much larger (e.g., ref. [47]), yet the magnitude of African hydrological change was similar (Fig. 3). This might be taken suggest that high- and mid-latitude temperature only played a secondary role in controlling African precipitation compared to local biogeophysical feedbacks. However, other factors including ice sheet retreat and tropical warming likely had an effect on African precipitation at the AHP onset, inhibiting a direct comparison. Nonetheless, it seems likely[8, 42, 43] that vegetation, dust, lake and wetland feedbacks played a role in amplifying the hydroclimatic shifts at the AHP termination.

In summary, our findings suggest that the effect of rapid high- and mid-latitude temperature changes on tropical African hydroclimate was not restricted to the glacial and deglacial, but also played a decisive role in triggering the AHP termination. Teleconnection of high-mid latitude temperatures with the TEJ reduced JJA precipitation in the Sahel-Sahara, tipping the hydrological system towards an arid state. Although the high-latitude temperature changes were relatively small during the Holocene, the associated initial drying was the required trigger for vegetation, soil moisture, dust and lake feedbacks that together resulted in a large and rapid aridification. From these findings, it appears that future changes in high-latitude SST, in particular associated with sea-ice changes, may have strong implications for low-latitude hydroclimate[66].

## Methods

**Sediment core and age model**. Marine sediment core GeoB4905-4 was recovered at 2°30.0′ N, 09°23.4′ E from 1328 m water depth offshore Cameroon[67]. The age model of the core is based on 12 radiocarbon ages[68, 69] that have been re-calibrated using the Marine13 curve with a reservoir age of 0.4 ± 0.1 kyr. The age-depth relationship was constructed using the software BACON 2.2[70] and represents the median of 10,000 iterations (Supplementary Fig. 6). The mean age uncertainty (1σ) over the last 25 ka is ± 0.3 kyr.

**n-Alkane extraction and purification**. Extraction and purification were performed at MARUM—Center for Marine Environmental Sciences, Bremen. Sediment samples of 10 ml were taken from core GeoB4905-4 with syringes, which yielded up to 9 g of dry sediment. Samples were oven dried at 40 °C, homogenised and squalane internal standard was added before extraction. Organic compounds were extracted with a DIONEX Accelerated Solvent Extractor (ASE 200) at 100 °C and 1000 psi using a 9:1 mixture of dichloromethane to methanol for 5 min, which was repeated three times. The saturated hydrocarbon fraction was obtained by elution of the dried lipid extract with hexane over a silica gel column (mesh size 60)

followed by elution with hexane over AgNO$_3$-coated silica to remove unsaturated hydrocarbons.

**Isotopic analyses**. Isotopic analyses were performed at MARUM—Center for Marine Environmental Sciences, Bremen. n-Alkane $\delta^{13}$C analyses were carried out using a ThermoFisher Scientific Trace GC Ultra coupled to a Finnigan MAT 252 isotope ratio monitoring mass spectrometer via a combustion interface operated at 1000 °C. Isotope values were calibrated against external $CO_2$ reference gas. The squalane internal standard yielded an accuracy of 0.4‰ and a precision of 0.2‰ (n = 371). Samples were run at least in duplicate, with a reproducibility of on average 0.1‰ for the $C_{29}$ n-alkane. $\delta$D values of n-alkanes were measured using a ThermoFisher Scientific Trace GC coupled via a pyrolysis reactor operated at 1420 °C to a ThermoFisher MAT 253 isotope ratio mass spectrometer (GC/IR-MS). $\delta$D values were calibrated against external $H_2$ reference gas. The squalane internal standard yielded an accuracy of 1‰ and a precision of 3‰ on average (n = 428). Samples were analysed at least in duplicate, with an average reproducibility of 1‰ for the $C_{29}$ n-alkane. Repeated analysis of an external n-alkane standard between samples yielded a root-mean-squared accuracy of 2‰ and a standard deviation of on average 3‰. The H$_3$-factor had a mean of 6.00 ± 0.02 and varied between 5.83 and 6.19 throughout analyses.

**$\delta$D$_{wax}$ adjustments**. We adjusted $\delta$D$_{wax}$ for ice volume (following e.g., ref. [4]) using a seawater $\delta^{18}$O curve[71] and converting to $\delta$D assuming a Last Glacial Maximum (LGM) increase of 7.2‰ (Fig. 3a). We use 7.2‰ rather than 8‰ because sediment pore water $\delta^{18}$O and $\delta$D measurements[72] suggest that the glacial $\delta$D increase has a mean value of 7.2‰. We also adjusted the $\delta$D$_{wax}$ record for vegetation changes (e.g., ref. [73]) using published fractionation factors (−123‰ ± 31‰ for C$_3$ trees, −139‰ ± 27‰ for C$_4$ grasses; ref. [18]). End-member $C_{29}$ $\delta^{13}$C$_{wax}$ values used for C$_3$ and C$_4$ vegetation were −35.7‰ and −21.4‰, respectively. The large uncertainties reflect different physiology, water source and seasonal timing of synthesis between plant types. This in turn highlights that a vegetation adjustment distinguishing only between C$_3$ and C$_4$ may not capture all potential vegetation changes, for example, between the input of shrubs, bushes and forbs that constitute a small fraction of the source areas[74]. Nonetheless, the highly integrated signal in marine sediments likely averages out much of the vegetation-type effect on $\delta$D$_{wax}$, suggesting such an adjustment to be appropriate in this instance. Overall, the vegetation and ice-volume adjusted $\delta$D$_{wax}$ record (Fig. 3b) is similar to the unadjusted record (Fig. 3a), highlighting that the adjustments have a minor effect on the climate signal. Although $\delta$D$_{wax}$ records are sometimes adjusted for temperature[75], it is difficult to estimate the past relationship between temperature and $\delta$D$_p$. Given that the sea surface temperature record from GeoB4905-4[69] evolved similarly but in antiphase to $\delta$D$_{wax}$, a temperature adjustment would act to enhance the magnitude of past $\delta$D$_{wax}$ changes, suggesting the estimated magnitude of past $\delta$D$_p$ changes to be conservative.

**SiZer analysis**. In order to assess the timing and significance of the transitions in our $\delta$D$_{wax}$ record, we performed a SiZer (Significant Zero crossings of derivatives) analysis[76]. This creates a family of Gaussian smooths for the data, and for each smooth identifies the time periods during which the derivative is significantly different from zero. To compare the rapidity of the transitions, we calculated the mean rate of change for these identified time periods.

**Climate modelling**. Investigations of the effect of high-latitude cooling on African hydroclimate were performed with simulations of a high-resolution version of the fully coupled Community Climate System Model version 3 (CCSM3). In this model version, the atmosphere model has a T85 (1.4° transform grid) resolution with 26 levels in the vertical, while the ocean has a nominal 1° horizontal resolution with 25 levels[77]. To study AHP conditions, we analysed a control simulation at 8.5 ka. In this early Holocene (EH) experiment, we used the orbital parameters and green-house gas concentrations for 8500 years before present (CO$_2$ = 260 ppmv, CH$_4$ = 660 ppbv, N$_2$O = 260 ppbv)[78]. The EH experiment has been spun up over a period of 1400 years. In order to cool down the northern extratropics, a freshwater hosing was subsequently applied to the EH control run (experiment EH$_{fre}$), in which freshwater at a rate of 0.2 Sv was injected into the northern North Atlantic for 400 years[79]. From both experiments (control and hosing) the last 100 years were taken and averaged for analyzes.

Investigations of the source of the atmospheric $\delta$D$_p$ signal were performed with a transient run of the intermediate complexity isotope-enabled climate model iLOVECLIM[80–82]. We studied the last 25 kyr of a 150 kyr simulation, which was run with the atmosphere at 5.6° resolution and used accelerated forcing (irradiance, GHGs and ice sheets were updated with an acceleration factor 10)[83]. Intermediate complexity models such as this have difficulty reproducing precipitation, but have the advantage of producing a continuous transient simulation of water isotopes for comparison with proxy data.

**Code availability**. CCSM3 source code is disseminated via the Earth System Grid (www.earthsystemgrid.org). Full model documentation is available at http://www.cesm.ucar.edu/models/ccsm3.0/.

The *iLOVECLIM* source code is based on the LOVECLIM model version 1.2, whose code is accessible at http://www.elic.ucl.ac.be/modx/elic/index.php?id=289. The developments on the *iLOVECLIM* source code are hosted at https://forge.ipsl.jussieu.fr/ludus, but are not publicly available due to copyright restrictions. Access can be granted on demand by request to D. M. Roche (didier.roche@lsce.ipsl.fr) to those who conduct research in collaboration with the *iLOVECLIM* users group.

**Data availability**. The datasets generated during the current study are available in the PANGAEA repository https://doi.pangaea.de/10.1594/PANGAEA.880119.

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

## Acknowledgements

J.A.C. was supported by the Helmholtz Postdoc Programme (PD-001) and the Alfred Wegener Institute Helmholtz Center for Polar and Marine Research, Bremerhaven. E.S., B.B. and M.P. were supported by the German Science Foundation (DFG) within the Priority Programme (SPP) 1266"Interdynamic" (Sche903/9; PR1050/4) and the DFG Research Center/Cluster of Excellence "The Ocean in the Earth System" at MARUM–Center for Environmental Sciences. T.C. and D.R. are supported by CNRS-INSU. C.S. was partly supported by a Grant from the French government through Agence Nationale de la Recherche (ANR) under the 'Investissements d'Avenir' programme, reference ANR-10-LABX-19-0. CCSM3 model experiments were performed on the HLRN supercomputer. We thank Raquel Nieto for her assistance with the FLEX-PART computations. J.A.C. is grateful to Yannick Garcin and Jule Müller for discussion.

## Author contributions

J.A.C. designed the study, analysed the data and wrote the manuscript. B.B. performed lipid extraction and purification, and E.S. designed the study and performed the isotopic analyses. M.P. performed the CCSM3 model analysis. T.C. and D.R. provided the *i*LOVECLIM model output. S.M. assisted with the BACON analysis. L.G. provided the FLEXPART output. All authors contributed to the discussion and interpretation.

## Additional information

**Competing interests:** The authors declare no competing financial interests.

