## [Peer Review File · Nature Communications]

Reviewers' comments:

Reviewer #1 (Remarks to the Author):

In the manuscript, "Rapid termination of the African Humid Period triggered by northern high-latitude cooling Rapid termination African Humid Period," Collins et al. showed that the leaf wax δD from Gulf of Guinea presents a sign of rapid termination of the humid period. They argued that the abrupt termination of the humid period is prevalent around the Africa, and suggested that the humid period may have ended because of the northern high-latitude cooling. Northern high-latitude cooling could influence the tropical jets and eventually dry the area. This hypothesis was tested using the results from NCAR CESM runs (Liu et al., 2014; Figure S6), although the intermediate-complexity model did not completely support their claim.

The hypothesis is very interesting, was delivered carefully, and is partly backed up by the climate modeling. It will interest a wide range of audience. I recommend the publication of this paper after my comments below are addressed.

I'd like to mention that this review is mostly about the presentation of the hypothesis, the interpretation of isotopes, and the model results; the justification related to treating the proxy record is beyond my expertise.

1. Line 148: Figure 1c is an important figure, but it is very small and difficult to see. Also, the authors claimed that "the Douala and N'djamena GNIP stations exhibit similar long-term mean δD_p values (Fig. 1c)." Without any analysis and from Fig 1c alone, it is difficult to reach this conclusion.

2. Line 150, Figure S4: Precipitation in Sahel does not show any abrupt change, and actually precipitation in Cameroon is not even decreasing after 5.5k years ago, although δD shows a strong increase, similar to the proxy data. I don't understand how the authors could conclude that the precipitation change around 5.5K is abrupt. I understand that this is the intermediate-complexity GCM (its atmospheric component is particularly weak), so the response may not be correct. However, the authors did not explain this discrepancy mechanistically (aside from integrating the upstream region) or the limitations of the intermediate-complexity GCM.

3. The authors emphasized the role of tropical jets in modulating the hydrological cycle of Africa. I understand that Figure S6 is taken from Liu et al. (2014). Would it be possible to at least address how the cooling of North Atlantic influences tropical jets? The authors explanation on "a cool summer temperature anomaly over the northern Atlantic and northern Africa (Fig. S6a) can reduce the meridional gradient of mean tropospheric temperature and hence, reduce the speed of the TEJ" is confusing. How does the decrease of high latitude temperature reduce the meridional gradient of mean tropospheric temperature?

Reviewer #2 (Remarks to the Author):

The central claim of the paper submitted by Collins et al. is that the Northern Hemisphere cooled at 5.5ka, which weakened windspeeds of the TEJ, which abruptly decreased precipitation across much of Africa (north of 10 degrees S), and that this abrupt decrease was the 'tipping point' toward a more arid state that was then reinforced by feedbacks with vegetation, soil moisture, and SSTs. It is a nice explanation and it is certainly plausible. The record itself is very interesting. However, while there is no one major hole in their argument, there are many small holes that, added together, make me question whether the evidence is compelling enough to support their claim. Without compelling evidence I do not believe it is the right fit for a broad-reaching journal such as Nature Communications.

However if the authors can provide more convincing evidence, then perhaps the issue can be remedied. That evidence would especially need to address:

I) Improved discussion of the controls on dDp. Clear discussion of moisture sources for dDwax in the Gulf of Guinea vs. Lake Bosumtwi and the other dDwax records from Africa. Additionally, a shift to prioritization of dDwax records in the discussion and in the figures, so as not to confuse the reader with “apples to oranges” comparisons between dDwax and terrigenous flux or lake level, and so as to clarify the mechanisms connecting the North Atlantic with dDp in different parts of Africa. See my comments #4 and #5b below.

II) Additional evidence to support the “abruptness” of the shift at 5.5ka, considering the change in resolution of the dataset around that time. Additional evidence could include improved resolution in the late Holocene, a “rate of change” analysis performed on synthetic data, or both.

III) A clarified discussion of the mechanism that connects NH SST to the TEJ under mid-Holocene conditions, possibly through additional analyses of model simulations presented in the supplement.

My comments are below, separated by each component of the authors’ central claim.

1) The Northern Hemisphere cooled at 5.5ka.

- This is well-argued in the paper.

2) Such cooling weakened windspeeds along the Tropical Easterly Jet via an atmospheric mechanism.

- This mechanism seems possible, but could use additional explanation. The mechanism cited to explain the Sahelian drought in the 1960’s, discussed by Liu and Chiang (2012, J Clim), focuses much more on the increase in SLP, with weakening of the TEJ as the final straw. Would high pressure over the Sahel/Sahara matter as much, or respond as rapidly, in an early/mid-Holocene world with more vegetation and overall moisture? The authors should provide more detail into how this mechanism would actually work in the mid-Holocene. It seems they could use their model simulation to perform at least a basic test of connections between NH SST and North African SLP.

- Patricola and Cook (ref 11) show that weakening of the TEJ is a possible cause of abrupt climate changes, but they cite vegetation and soil moisture feedbacks as a necessary intermediary. Cook (J. Climate, 1999) also showed that shifts in the AEJ also rely on soil moisture gradients. It seems unlikely that NH cooling could weaken the TEJ alone, without invoking feedbacks with vegetation or at least soil moisture. Therefore, it seems plausible that gradually changing vegetation (in response to precession) could have served as its own aridity triggering, without needing to involve cooling in the Northern Hemisphere.

3) Weakened windspeeds led to decreased precipitation along the TEJ.

- There is good evidence that the TEJ and AEJ strengthen during West African monsoon season. This is well argued and well cited in the paper.

4) Decreased precipitation was widespread, across most of Africa north of 10S.

This is the crux of the paper. In order for the “NH cooling→ TEJ weakening→ Initial abrupt tilt

towards aridity" theory to work, the abrupt aridity needs to be widespread, because the position and strength of the TEJ and AEJ help determine the locus and strength of precipitation across the tropical rain belt, AND because TEJ strength is strongly associated with strong West African monsoons. Therefore, the West African Monsoon region as well as parts of central and Eastern Africa would need to show abrupt decreases in precipitation.

I do not think the authors can make the case here. My first concern is that the records discussed in lines 167-212 including those shown in Figure 4d,e,f are not really apples-to-apples comparisons with dDwax. As the authors explain (see 5b below), dDwax reflects precipitation and humidity patterns on a regional scale, not a local scale. However, proxies like terrigenous dust flux and lake level are very different, and reflect strong changes in surface evaporation, vegetation cover, cloud cover, winds, etc etc. They are far more likely to be local signals than dDwax. (Moreover, the lake level records in Figure 4 are so low-resolution that it seems odd to include them on the figure, when high resolution records of dDwax itself are available from many sites.) I understand the desire to simplify some of these proxies by just saying they reflect "hydrology" (eg line 201 when describing the terrigenous flux record). In some cases those simplifications are justified. However in this case, the nuances of how each proxy is interpreted are very important, because they determine ways in which truly abrupt changes could be recorded as gradual, and vice versa.

My second concern is that, if dDp is truly a proxy for regional precipitation, then how is it possible that the Bosumtwi record is so different from this one? The authors say that the Bosumtwi dDwax record probably just reflects local changes in climate, but that argument in itself is contradictory to the authors' claims that dDp reflects regional processes. Considering what we know about dDwax it is no surprise that the Bosumtwi dDwax record is different from the Bosumtwi lake level record. As I state above, they are proxies for very different things.

The most convincing aspect of this section of the paper, to me, is the similarity between the Gulf of Guinea record and Gulf of Aden records. This should be expanded, focusing on dDwax records for like-with-like comparisons. Figure 5 in Berke et al., 2012, QSR shows a nice compilation of many dDwax records from East Africa, all of which should respond to changes in the TEJ. Only a couple of them (Tanganyika, though resolution is very limited) show an abrupt shift around 5ka. The rest show gradual changes or a structure that looks totally different.

There are obvious reasons for differences among dDwax records, the most obvious of which are differences in rainfall seasonality and differences in moisture source. The discussion of moisture source (Figure 3) could be relied on more heavily here to aid the interpretation. The discussion in lines 141-156 is intriguing but falls short of declaring exactly what moisture sources the GoG dDwax DOES reflect. Is it mostly Sahel? SW Africa? I suggest that the authors use this part of their paper to strengthen their interpretation of dDwax and therefore dDp. Yes, dDp is a regional recorder, but be more specific. What region? What time of year? What processes are most important—convection over the Congo basin? Southwesterly moisture transport onto the African continent? How do these sources differ from those of Bosumtwi? How do they differ from the Gulf of Aden? What can explain the similarities?

Perhaps the argument becomes that the Gulf of Guinea's moisture sources cause it to pick up a Sahel/Sahara like response rather than a deep tropical response, explaining the similarities to the Aden record and the dissimilarities to the Bosumtwi record. This would be intriguing. It would also point to a more specific season and perhaps a more specific mechanism for the NH cooling/TEJ interaction, and would relax the need for the authors, or anyone, to find one record that can explain nearly all of Africa.

5) The Gulf of Guinea dDwax record reflects these widespread precipitation decreases because:

5a) dD_{wax} records dD_p with fidelity

- It makes sense that dD_{wax} would not be complicated by major changes in vegetation. The d¹³C changes are only a few permil and that should only account for a few permil of change in dD. However, some of the changes that the authors discuss, for example the LGM-present isotopic difference, are small enough that they could be impacted by vegetation. So while I believe the authors that vegetation is not important, I do not think they have technically offered enough evidence to back up their claim. Therefore:

- The vegetation corrected should be explicitly shown in a figure. Please revise Figure 2 to include the original dD_{c29}, ice-volume corrected dD_{c29}, and vegetation-corrected dD_p. Figure S2 can be kept as-is.

- Additionally, the vegetation correction should be clearly explained in the "Methods" section of the paper. Otherwise, the reader is referred to the Supplement, which then refers to a study from 2013, which the reader then must find. In that study I cannot easily find the d¹³C endmembers for C3 and C4 used to calculate epsilon, as they go back to a previous study from 2011, and when I refer to that study, I am no longer clear whether these are the same compounds or chain-lengths. The vegetation correction methodology is not well established enough to ask non-specialists to chase down a paper like this, and it is controversial enough among organic geochemists that even specialists would prefer to see it stated explicitly.

- The most important thing is that there are no abrupt d¹³C_{wax} shifts at 5.5ka, so the dD_{wax} shifts are probably not caused by vegetation changes, at least not C3/C4 changes. This is an important point for the authors to make since the 5.5ka shift is the focus of this paper. Even if vegetation changes could explain some LGM-present differences, for example, they could not explain the abrupt shift at 5.5ka.

5b) dD_p generally reflects regional precipitation amount

- It is stated on line 105 that "dD_p is anticorrelated with convective precipitation amount and intensity." This is not supported and possibly misleading since a considerable amount of work has been done recently to show that the amount effect is stronger in tropical stratiform rain than convective rain. See Kurita et al., 2011 and 2013, both in JGR-Atmospheres; Conroy et al., 2016, JGR-Atmospheres; Aggarwal et al., 2016, Nature Geoscience. Perhaps this is a terminology issue, see my comment below about the way the term 'convection' is used throughout this paper.

However, neither does the cited Moerman et al paper support this claim. The main finding of that paper is that tropical dD_p is closely tied to regional scale changes in precipitation amount rather than local scale changes in precipitation amount. The explanation of the amount effect given in lines 106-108 seems to be based on theories of Dansgaard (1964) as well as the Risi papers. Both Dansgaard and Risi are describing processes that happen locally, whereas the Moerman mechanisms translate regional scale precipitation to local dD via "upstream effects" and changes in moisture source. These are very different from the local amount effect mechanisms discussed here.

I suggest doing away with this paragraph entirely. The authors describe the ways in which dD_p reflects regional scale patterns in the Discussion section, lines 131-156, and that is a more appropriate place for interpretation anyway.

- The authors claim that recycling is not important for their record, but I do not agree. Recycled moisture from places as rainy as the Congo may be enriched relative to non-recycled moisture from the Congo, but still very depleted relative to other moisture sources to the Gulf of Guinea watershed/waxshed, in part from a strong amount effect and in part from strong distillation as it

travels from the Congo. See Costa et al., 2014, QSR as an example. That study claimed that AHP moisture was isotopically depleted beyond simple precipitation amount due to increased influx of Congo moisture. We could very well be seeing the same thing in this GoG record.

- Regarding moisture sources presented in Figure 3: If recycled moisture is indeed an important component, and there is ample evidence that it should be, then the deep equatorial Congo Basin could easily be a source of moisture even when $P \gg E$. These maps basically show a giant hole over the central Congo Basin (especially 3b, MAM) which does not seem realistic considering the amount of moisture recycling that takes place there.

5c) Changes in dD_{wax} at 5.5ka are abrupt

To the eye, the change in dD_{wax} between 5-6 ka does seem abrupt, and I appreciate the authors' attempt to quantify the rate of change rather than just saying it looks that way. However, I am skeptical of the Rate of Change analysis, particularly as it pertains to the mid-Holocene. In the Supplement the authors admit that the resolution of the dD_{wax} record is too low in the late Holocene (after 5ka) to assess the end of the AHP termination. It is difficult to tell the exact mid-Holocene resolution from Figure 2 as the individual data points are not clearly shown. However the authors give a ~ 0.2 kyr resolution in the caption for Figure 2. If the resolution is 0.2kyr, the low resolution of the record begins around 5kyr, and rates of change are determined by a 5-point smooth, then how trustworthy is it that "the maximum rate of change during the Holocene was at 5.3ka" (lines 120-121)? Couldn't this change at 5.3ka just reflect a change in the resolution of the record after 5ka, rather than an actual increase in dD_{wax} RoC? Since the authors have pitched this paper as an answer to the "abrupt versus gradual" question, this point needs to be crystal clear. I suggest testing this "abrupt termination" assertion with additional analyses, either by increasing the resolution of the record in the late Holocene, or by testing the effects of resolution using synthetic datasets and the RoC code (or both). In addition, caveats should be explicitly stated in the main text.

5d) Changes in dD_{wax} at 5.5ka are of large magnitude (comparable to the YH and H1, both believed to have been driven by NH temperature)

- The SiZer method needs to be described in the main text, not in the Supplement, as it is key to the authors' interpretation of "large and significant" changes in dD_{wax} .

- The authors argue that the rate of change at 5.5ka is large, but from Figure S3 it appears that this large change only barely crosses the Ruppert-Sheather-Wand bandwidth.

Additional, line-by-line comments:

- There seems to be some confusion in the paper about the designation of "convective precipitation." The authors take care to point out "convective" precipitation, for example lines 61 and 107. It's unclear whether the authors just mean precipitation in general, or if they truly only mean convective precipitation (rather than stratiform, which is also present in central Africa for example in Mesoscale Convective Systems). If they mean convective precipitation, are they referring to deep or shallow? These distinctions are important when discussing the mechanisms for windspeeds in the TEJ and AEJ influencing precipitation. If the authors simply mean precipitation in general then they should remove these references to convection, for the sake of the atmospheric convection researchers who read Nat Comms.

Line 77: The age model is not in fact 'published' but is just given in the Supplement (unless there is a missing reference in the Supplement). Providing the age model in the Supplement is OK, but in the main text please give the basics: What was used to date the core, how many radiocarbon dates, the general age uncertainty, and the age uncertainty around the 5.5ka event in particular.

Line 83: Why are these C3 and C4 end members chosen, rather than using the more updated compilation in Sachse et al. 2012?

Line 88: Please provide the $\delta^{13}\text{C}_{\text{C29}}$ values that you cite as evidence for C4-dominated vegetation in the Sahara-Sahel. Also, for the sake of comparability, I strongly recommend using real measurements of plants or sediments rather than the model/reanalysis dataset cited here.

Lines 214-219: The authors might consider CO_2 as a possible forcing, rather than, or in addition to, precession. Nonlinear responses to CO_2 forcing are very plausible.

Reviewer #3 (Remarks to the Author):

This paper is well motivated and suggests an interesting additional possible mechanism to explain the abrupt AHP termination. One of the main questions I have is whether the paleo-proxy is representative of the West African monsoon and Sahel vegetation. Following justification of this point, and some others below, I would consider recommending this manuscript for publication.

Major comments:

1. This study uses a marine core located in the Gulf of Guinea. It appears that the primary sources of terrestrial material to the core are from equatorial Africa. It is not clear how a core with these sources would represent the monsoon and Sahelian vegetation at ~15N, which can be driven by different physical mechanisms than equatorial precipitation. A clearer/stronger justification for the use of this paleo-proxy cite is needed in this regard.
2. Previous studies have suggested that vegetation-atmosphere feedbacks could have served as a mechanism for the abrupt AHP termination. This study puts forth another possible explanation -- that the initial precipitation decrease leading out of the AHP was driven by northern high-latitude SST cooling that resulted in a teleconnected TEJ response. Some of the wording is too strong given the lack of direct modeling evidence (e.g., line 243: “likely”). I am not necessarily recommending adding a climate modeling component to the study (although it would be interesting!), but the wording should be softened (“could” or “possible”) unless climate model experiments are added. In addition, I suggest balancing the abstract by including a mention of the role of vegetation-atmosphere feedbacks, as it remains unclear whether the abrupt termination was driven by high-latitude cooling, vegetation feedbacks, or both. Furthermore, paleo-proxy data suggest the onset of the AHP was also abrupt (e.g., deMenocal et al. 2000). Does the high-latitude SST mechanism explain the abrupt AHP onset as well?
3. Line 47: Some paleo records show an abrupt AHP onset/termination, while others don't. Is there a reason to believe one conclusion vs. the other? Are the records that show a gradual AHP termination anomalies among the greater set? A few words to put this in perspective would help.
4. Line 48: Similar comment regarding the coupled models. Are there modeling simulations that suggest that biogeophysical feedbacks could have caused the vegetation collapse? Two studies come to mind – one that simulates an abrupt AHP termination, and another that simulates a gradual one (Claussen et al, 1999; Wang et al. 2005). There may be others as well.

- Claussen, M., C. Kubatzki, V. Brovkin, and A. Ganopolski (1999), Simulation of an abrupt change in Saharan vegetation in the mid-Holocene, *Geophys. Res. Lett.*, 26(14), 2037– 2040, doi:10.1029/1999GL900494.

- Wang, Y., L. A. Mysak, Z. Wang, and V. Brovkin (2005), The greening of the McGill Paleoclimate Model. Part II: Simulation of Holocene millennial-scale natural climate changes, *Clim. Dyn.*, 24, 481 – 496, doi:10.1007/s00382-004-0516-8.

We would like to thank the reviewers for their very extensive and constructive comments. We have now significantly revised the manuscript taking all of these comments into account, including additional leaf-wax data and model experiments. Point-by-point responses are described below (in bold) and changes to the manuscript are marked with an arrow.

1) *As you will see from the reports, the major issues raised concern discrepancies among the proxy records*

- **We have now only included δD_{wax} records in our main comparison (Fig. 5). Using additional moisture source data we have developed robust arguments for the similarities and discrepancies between the records. Please see comments 10-15 for a detailed description.**

2) *and the support provided for your proposed mechanism. It would seem that the addition of a modelling component would certainly prove beneficial in supporting the latter.*

- **We have now added an additional modelling component to the manuscript to illustrate our proposed mechanism. Please see comments 5, 7 and 28 for a detailed description.**

We hope you will find the referees' comments useful as you decide how to proceed. Should further experimental data or analysis allow you to address these criticisms, we would be happy to look at a substantially revised manuscript. However, please bear in mind that we will be reluctant to approach the referees again in the absence of major revisions. If the revision process takes significantly longer than 6 months, we will be happy to reconsider your paper at a later date, as long as nothing similar has been accepted for publication at Nature Communications or published elsewhere in the meantime.

Reviewer #1 (Remarks to the Author):

In the manuscript, "Rapid termination of the African Humid Period triggered by northern high-latitude cooling Rapid termination African Humid Period," Collins et al. showed that the leaf wax dD from Gulf of Guinea presents a sign of rapid termination of the humid period. They argued that the abrupt termination of the humid period is prevalent around the Africa, and suggested that the humid period may have ended because of the northern high-latitude cooling. Northern high-latitude cooling could influence the tropical jets and eventually dry the area. This hypothesis was tested using the results from NCAR CESM runs (Liu et al., 2014; Figure S6), although the intermediate-complexity model did not completely support their claim.

The hypothesis is very interesting, was delivered carefully, and is partly backed up by the climate modeling. It will interest a wide range of audience. I recommend the publication of this paper after my comments below are addressed.

I'd like to mention that this review is mostly about the presentation of the hypothesis, the interpretation of isotopes, and the model results; the justification related to treating the proxy record is beyond my expertise.

3). *Line 148: Figure 1c is an important figure, but it is very small and difficult to see. Also, the authors claimed that "the Douala and N'djamena GNIP stations exhibit similar long-term mean δD_p values (Fig. 1c)." Without any analysis and from Fig 1c alone, it is difficult to reach this conclusion.*

Agreed. This was based on the long-term mean values from the IAEA datasets.

- **We have included the long-term mean δD_p values in the main text (lines 159-161). We have also now enlarged Figure 1c to highlight not only the**

rainfall regime, but also the steeper precipitation- δD_P relationship in the Sahel compared to Cameroon.

4). Line 150, Figure S4: Precipitation in Sahel does not show any abrupt change, and actually precipitation in Cameroon is not even decreasing after 5.5k years ago, although dD shows a strong increase, similar to the proxy data. I don't understand how the authors could conclude that the precipitation change around 5.5K is abrupt. I understand that this is the intermediate-complexity GCM (its atmospheric component is particularly weak), so the response may not be correct. However, the authors did not explain this discrepancy mechanistically (aside from integrating the upstream region) or the limitations of the intermediate-complexity GCM.

Agreed - this was unclear in the original version. Our conclusion of a rapid hydrological change was mainly based on the δD_{wax} proxy data. The intermediate complexity model data was included to highlight that the Sahel was a partial source of the δD_P signal in Cameroon. Even though the magnitude of precipitation changes may not be accurate in the intermediate complexity model, it is nonetheless useful for illustrating a non-local control on the precipitation isotopes. A possible reason for the absence of a rapid precipitation decrease at 5.8-4.7 ka may be that an accelerated forcing technique was used for the simulation and that relevant vegetation, soil moisture and dust feedbacks are likely not adequately accounted for in the model. Moreover, changes at the HS1 and YD terminations are absent likely because freshwater fluxes induced by ice-sheet collapses are not included in the transient simulation.

- **We have now clarified our reasons for using the intermediate complexity model (line 163-165). We also now include a detailed description of the limitations of the intermediate complexity model and possible reasons that it does not simulate an abrupt response (Supplementary Material lines 68-75).**

5a) *The authors emphasized the role of tropical jets in modulating the hydrological cycle of Africa. I understand that Figure S6 is taken from Liu et al. (2014). Would it be possible to at least address how the cooling of North Atlantic influences tropical jets?*

The cooling of the North Atlantic causes cooling of the troposphere over Africa, reducing the temperature gradient in Africa and, by the thermal wind relationship, reducing the TEJ speed.

- **We now illustrate the effect of cooling on the TEJ using new model data: a freshwater forcing experiment on an early Holocene high-resolution run of the fully coupled CCSM3 climate model (Fig. 7). We include a detailed description of the mechanism in the manuscript (lines 319-349).**

5b) *The authors explanation on "a cool summer temperature anomaly over the northern Atlantic and northern Africa (Fig. S6a) can reduce the meridional gradient of mean tropospheric temperature and hence, reduce the speed of the TEJ" is confusing. How does the decrease of high latitude temperature reduce the meridional gradient of mean tropospheric temperature?*

Agreed - this was not clear. The cooling anomaly spreads south towards Africa. Thus, cooling in the Sahara reduces the meridional temperature gradient between the Sahara and the Gulf of Guinea.

- **This has been clarified (lines 329-332)**

Reviewer #2 (Remarks to the Author):

The central claim of the paper submitted by Collins et al. is that the Northern Hemisphere

cooled at 5.5ka, which weakened windspeeds of the TEJ, which abruptly decreased precipitation across much of Africa (north of 10 degrees S), and that this abrupt decrease was the 'tipping point' toward a more arid state that was then reinforced by feedbacks with vegetation, soil moisture, and SSTs. It is a nice explanation and it is certainly plausible. The record itself is very interesting. However, while there is no one major hole in their argument, there are many small holes that, added together, make me question whether the evidence is compelling enough to support their claim. Without compelling evidence I do not believe it is the right fit for a broad-reaching journal such as Nature Communications.

However if the authors can provide more convincing evidence, then perhaps the issue can be remedied. That evidence would especially need to address:

- Improved discussion of the controls on dDp. Clear discussion of moisture sources for dDwax in the Gulf of Guinea vs. Lake Bosumtwi and the other dDwax records from Africa. **(see comments 10-19).**
- Additionally, a shift to prioritization of dDwax records in the discussion and in the figures, so as not to confuse the reader with "apples to oranges" comparisons between dDwax and terrigenous flux or lake level, and so as to clarify the mechanisms connecting the North Atlantic with dDp in different parts of Africa. See my comments #4 and #5b below. **(see comment 13).**
- Additional evidence to support the "abruptness" of the shift at 5.5ka, considering the change in resolution of the dataset around that time. Additional evidence could include improved resolution in the late Holocene, a "rate of change" analysis performed on synthetic data, or both. **(see comment 20).**
- A clarified discussion of the mechanism that connects NH SST to the TEJ under mid-Holocene conditions, possibly through additional analyses of model simulations presented in the supplement. **(see comments 7 and 8).**

My comments are below, separated by each component of the authors' central claim.

6) The Northern Hemisphere cooled at 5.5ka - This is well-argued in the paper.

Thank you

7) Such cooling weakened windspeeds along the Tropical Easterly Jet via an atmospheric mechanism. This mechanism seems possible, but could use additional explanation. The mechanism cited to explain the Sahelian drought in the 1960's, discussed by Liu and Chiang (2012, J Clim), focuses much more on the increase in SLP, with weakening of the TEJ as the final straw. Would high pressure over the Sahel/Sahara matter as much, or respond as rapidly, in an early/mid-Holocene world with more vegetation and overall moisture? The authors should provide more detail into how this mechanism would actually work in the mid-Holocene. It seems they could use their model simulation to perform at least a basic test of connections between NH SST and North African SLP.

Agreed. As mentioned above, we tested the mechanism using the high-resolution CCSM3 model. The model shows that, even during the wetter conditions of the early Holocene, a summer temperature anomaly does indeed cause a significant weakening of the TEJ, reducing mid-upper tropospheric uplift, which reduces precipitation. In addition, a weakening of the Saharan Heat Low, as described in Liu and Chiang (2012) and Liu et al. (2014), reduces the moist low-level monsoonal inflow and hence moisture convergence.

- **The new model results are described in lines 319-349 and shown in Fig. 7.**

8a) Patricola and Cook (ref 11) show that weakening of the TEJ is a possible cause of

abrupt climate changes, but they cite vegetation and soil moisture feedbacks as a necessary intermediary. Cook (J. Climate, 1999) also showed that shifts in the AEJ also rely on soil moisture gradients.

Patricola and Cook (ref 11) and Cook (1999) both describe the response of the AEJ rather than TEJ. Nonetheless, based on these studies, it seems possible that the AEJ also acted as an additional feedback.

- **We now emphasise the potential feedback between hydroclimate and the AEJ (lines 351-354).**

8b) It seems unlikely that NH cooling could weaken the TEJ alone, without invoking feedbacks with vegetation or at least soil moisture.

Agreed. It is unlikely NH cooling was the sole driver - rather, we suggest it was the trigger of the feedbacks.

- **We now emphasise in the abstract and discussion that vegetation, soil moisture and dust feedbacks likely played a role in amplifying the climatic response (lines 32-34, 270-271, 360-363).**

8c) Therefore, it seems plausible that gradually changing vegetation (in response to precession) could have served as its own aridity triggering, without needing to involve cooling in the Northern Hemisphere.

If gradually changing vegetation were to trigger an abrupt response of the TEJ, this would require evidence for gradually decreasing precipitation between ca. 9 ka and 5.5 ka (in response to precession). Although there is slight evidence in our data, there is little evidence in the Gulf of Aden data for a gradual change in hydrology from about 9 ka. Moreover, the Patricola and Cook study was focussed on the AEJ in West Africa, and thus their results would not directly support a change in the TEJ (i.e. over central and eastern Sahel-Sahara). Finally, given that we see synchronous response at two widely spaced sites, this would seem to befit an extratropical trigger rather than vegetation, which may have been a more locally variable response.

9) Weakened windspeeds led to decreased precipitation along the TEJ.

- There is good evidence that the TEJ and AEJ strengthen during West African monsoon season. This is well argued and well cited in the paper.

Thank you

10) Decreased precipitation was widespread, across most of Africa north of 10S.

This is the crux of the paper. In order for the "NH cooling → TEJ weakening → Initial abrupt tilt towards aridity" theory to work, the abrupt aridity needs to be widespread, because the position and strength of the TEJ and AEJ help determine the locus and strength of precipitation across the tropical rain belt, AND because TEJ strength is strongly associated with strong West African monsoons. Therefore, the West African Monsoon region as well as parts of central and Eastern Africa would need to show abrupt decreases in precipitation. I do not think the authors can make the case here.

Agreed - this is an important question. As suggested in comments 11-15 below, and as shown in the CCSM3 model experiments, the precipitation changes were likely restricted to the Sahel-Sahara during JJA. We also chose alkenone temperature proxies (Fig. 6), because of their summer bias. As suggested by the reviewer, restriction to JJA relaxes the need for a continent-wide aridification (see comments 13-15). In terms of the West African Monsoon region, we have made some suggestions for the discrepancies with the Lake Bosumtwi record (see comment 12).

11) My first concern is that the records discussed in lines 167-212 including those shown in

Figure 4d,e,f are not really apples-to-apples comparisons with dD_{wax} . As the authors explain (see 5b below), dD_{wax} reflects precipitation and humidity patterns on a regional scale, not a local scale. However, proxies like terrigenous dust flux and lake level are very different, and reflect strong changes in surface evaporation, vegetation cover, cloud cover, winds, etc etc. They are far more likely to be local signals than dD_{wax} . (Moreover, the lake level records in Figure 4 are so low-resolution that it seems odd to include them on the figure, when high resolution records of dD_{wax} itself are available from many sites.) I understand the desire to simplify some of these proxies by just saying they reflect "hydrology" (eg line 201 when describing the terrigenous flux record). In some cases those simplifications are justified. However in this case, the nuances of how each proxy is interpreted are very important, because they determine ways in which truly abrupt changes could be recorded as gradual, and vice versa.

Agreed.

- **We have removed the comparison with dust and lake level from Fig. 5 (originally Fig. 4) and now only describe these examples in the text. We emphasise the difference between dust and hydrology and that the proxy significance might affect the respective response (line 252-254). In our main comparison figure (Fig. 5), we now only compare with the δD_{wax} record from the Gulf of Aden.**

12) *My second concern is that, if dD_p is truly a proxy for regional precipitation, then how is it possible that the Bosumtwi record is so different from this one? The authors say that the Bosumtwi dD_{wax} record probably just reflects local changes in climate, but that argument in itself is contradictory to the authors' claims that dD_p reflects regional processes. Considering what we know about dD_{wax} it is no surprise that the Bosumtwi dD_{wax} record is different from the Bosumtwi lake level record. As I state above, they are proxies for very different things.*

This is a good point. We now suggest there is likely to be a different cause, related to the source of the leaf waxes to the Lake Bosumtwi core site. As shown in previously published lake-level data, the lake level was 110m higher at 5.7 ka, and overflowed the crater rim. At this point, the catchment within the crater would have been relatively small. By about 1 ka the lake had declined to below present day levels (although the exact timing of the lake level decrease is not well constrained). The lake level decrease would have exposed sediments of the crater walls (likely deposited during the mid-Holocene), resulting in supply of pre-aged leaf waxes. Supply of pre-aged organic material is thought to explain spurious bulk OM radiocarbon ages during the later Holocene (Shanahan et al., 2008). Although the bulk OM age model appears to be robust between 5 ka to 3 ka, the bulk of the carbon for the age model is unlikely to be derived from leaf waxes, and so this does not rule out the input of pre-aged waxes. Interestingly, at 5.7ka the $\delta^{13}C_{wax}$ shows a large decrease of 8 per mil (more than twice the range of our entire dataset) within a few hundred years. This is a similar timing to changes in our record, although suggests a shift towards more tree vegetation rather than aridification. This might represent some evidence of a major change in leaf wax source - either due to erosion of the crater walls, and/or re-vegetation of the newly exposed crater walls, or changes in the relative contribution of dust.

- **A shortened version of the above argumentation is presented in lines 189-199.**

13) *The most convincing aspect of this section of the paper, to me, is the similarity between the Gulf of Guinea record and Gulf of Aden records. This should be expanded, focusing on dD_{wax} records for like-with-like comparisons. Figure 5 in Berke et al., 2012, QSR shows a nice compilation of many dD_{wax} records from East Africa, all of which should respond to changes in the TEJ. Only a couple of them (Tanganyika, though resolution is very limited)*

show an abrupt shift around 5ka. The rest show gradual changes or a structure that looks totally different.

We now mainly focus on comparing with the δD_{wax} record from the Gulf of Aden. We have further investigated moisture sources using the FLEXPART analysis (see comment 14). The analysis shows that the Sahel-Sahara hydroclimate signal and moisture source is the likely commonality of the two sites. Lake Victoria appears to show a gradual response, which might be because precipitation at Lake Victoria falls in the MAM season and is thus less likely to be affected by the Sahel-Sahara signal, which would be generated several months earlier during the JJA season.

- **We have now expanded the description of moisture sources to Cameroon (lines 87-96).**
- **We state that the Sahel-Sahara is a common moisture source to the Gulf of Guinea (lines 157-159, 215-218) and Gulf of Aden (lines 225-227). We suggest this might be a potential cause of the abruptness in Lake Tanganyika (line 232-233). We describe our argumentation for the absence of a rapid response at Lake Victoria in lines 235-240.**

14) *There are obvious reasons for differences among dD_{wax} records, the most obvious of which are differences in rainfall seasonality and differences in moisture source. The discussion of moisture source (Figure 3) could be relied on more heavily here to aid the interpretation. The discussion in lines 141-156 is intriguing but falls short of declaring exactly what moisture sources the GoG dD_{wax} DOES reflect. Is it mostly Sahel? SW Africa? I suggest that the authors use this part of their paper to strengthen their interpretation of dD_{wax} and therefore dDp . Yes, dDp is a regional recorder, but be more specific. What region? What time of year? What processes are most important—convection over the Congo basin? Southwesterly moisture transport onto the African continent? How do these sources differ from those of Bosumtwi? How do they differ from the Gulf of Aden? What can explain the similarities?*

The new detailed analysis of the moisture sources to the regions now uses both backward and forward analyses with the FLEXPART model (Fig. 2). The backward analyses highlight the moisture source region. The forward analysis highlights where this moisture falls as precipitation. We find that, averaged annually, 30% of the moisture falling in Cameroon is from the Sahel-Sahara, while 70% is from the SE Atlantic. For SON, (the main wet season in southern Cameroon and directly after the JJA season in the Sahel), this is even higher, with 40% from the Sahel-Sahara. The analysis also highlights that precipitation in the Gulf of Aden, and Lake Tanganyika is also partly derived from the Sahel-Sahara moisture source (Fig. 2).

- **The expanded FLEXPART analysis is described in lines 87 to 94 and shown in Fig. 2.**

15) *Perhaps the argument becomes that the Gulf of Guinea's moisture sources cause it to pick up a Sahel/Sahara like response rather than a deep tropical response, explaining the similarities to the Aden record and the dissimilarities to the Bosumtwi record. This would be intriguing. It would also point to a more specific season and perhaps a more specific mechanism for the NH cooling/TEJ interaction, and would relax the need for the authors, or anyone, to find one record that can explain nearly all of Africa.*

Agreed - this may be an additional mechanism to explain the discrepancy of our record with δD_{wax} from Lake Bosumtwi. FLEXPART shows that the Sahel-Sahara source for Bosumtwi is restricted to the western Sahel, rather than the central-east Sahel moisture source common to the Gulf of Aden and Cameroon. Lake Bosumtwi also receives most precipitation during MAM, rather than during SON as in southern Cameroon, and so may well be reflecting changes during a different

season. Nonetheless, we suggest that the supply of pre-aged material is the more likely explanation for the marked difference in the two records.

- We emphasise that the rapid aridification took place in the Sahel-Sahara during the summer season (JJA) (line 216). This is the same season as the northern hemisphere temperature anomalies (line 301-302).

16) *The Gulf of Guinea dD_{wax} record reflects these widespread precipitation decreases because: dD_{wax} records dD_p with fidelity. It makes sense that dD_{wax} would not be complicated by major changes in vegetation. The d13C changes are only a few permil and that should only account for a few permil of change in dD. However, some of the changes that the authors discuss, for example the LGM-present isotopic difference, are small enough that they could be impacted by vegetation. So while I believe the authors that vegetation is not important, I do not think they have technically offered enough evidence to back up their claim. Therefore:*

- *The vegetation corrected should be explicitly shown in a figure. Please revise Figure 2 to include the original dD_{c29}, ice-volume corrected dD_{c29}, and vegetation-corrected dD_p. Figure S2 can be kept as-is.*

- *Additionally, the vegetation correction should be clearly explained in the "Methods" section of the paper. Otherwise, the reader is referred to the Supplement, which then refers to a study from 2013, which the reader then must find. In that study I cannot easily find the d13C endmembers for C3 and C4 used to calculate epsilon, as they go back to a previous study from 2011, and when I refer to that study, I am no longer clear whether these are the same compounds or chain-lengths. The vegetation correction methodology is not well established enough to ask non-specialists to chase down a paper like this, and it is controversial enough among organic geochemists that even specialists would prefer to see it stated explicitly.*

Agreed. One additional point to note is that the temperature effect is likely minimising the changes in δD_p between the LGM and late Holocene. Given that the LGM was cooler, if we were to correct for this it would enhance the LGM-late Holocene day δD_{wax} difference. We do not correct for temperature, however, since we cannot reliably estimate the past δD_p -temperature relationship.

- **We have modified Fig. 3 (originally Fig. 2) to include the original δD_{wax} data, the ice volume corrected δD_{wax} , and the ice volume and vegetation corrected δD_{wax} . Fig. S2 has been left as is.**
- **We now describe the vegetation correction in the methods section (line 393-409).**
- **The potential temperature effect is briefly described in the methods section (line 411-414) and the likelihood that the LGM aridification is conservative is mentioned in the discussion (lines 172-174).**

17) *The most important thing is that there are no abrupt d13C_{wax} shifts at 5.5ka, so the dD_{wax} shifts are probably not caused by vegetation changes, at least not C3/C4 changes. This is an important point for the authors to make since the 5.5ka shift is the focus of this paper. Even if vegetation changes could explain some LGM-present differences, for example, they could not explain the abrupt shift at 5.5ka.*

Agreed.

- **We now include these points at the start of the discussion (line 150-152).**

18) *dD_p generally reflects regional precipitation amount*

- *It is stated on line 105 that "dD_p is anticorrelated with convective precipitation amount and intensity." This is not supported and possibly misleading since a considerable amount of work has been done recently to show that the amount effect is stronger in tropical stratiform rain than convective rain. See Kurita et al., 2011 and 2013, both in JGR-Atmospheres; Conroy et al., 2016, JGR-Atmospheres; Aggarwal et al., 2016, Nature*

Geoscience. Perhaps this is a terminology issue, see my comment below about the way the term 'convection' is used throughout this paper.

However, neither does the cited Moerman et al paper support this claim. The main finding of that paper is that tropical dDp is closely tied to regional scale changes in precipitation amount rather than local scale changes in precipitation amount. The explanation of the amount effect given in lines 106-108 seems to be based on theories of Dansgaard (1964) as well as the Risi papers. Both Dansgaard and Risi are describing processes that happen locally, whereas the Moerman mechanisms translate regional scale precipitation to local dD via "upstream effects" and changes in moisture source. These are very different from the local amount effect mechanisms discussed here.

I suggest doing away with this paragraph entirely. The authors describe the ways in which dDp reflects regional scale patterns in the Discussion section, lines 131-156, and that is a more appropriate place for interpretation anyway.

- **We have removed the paragraph in the introduction and we have changed 'convective precipitation' to 'precipitation' throughout.**

19) *The authors claim that recycling is not important for their record, but I do not agree. Recycled moisture from places as rainy as the Congo may be enriched relative to non-recycled moisture from the Congo, but still very depleted relative to other moisture sources to the Gulf of Guinea watershed/waxshed, in part from a strong amount effect and in part from strong distillation as it travels from the Congo. See Costa et al., 2014, QSR as an example. That study claimed that AHP moisture was isotopically depleted beyond simple precipitation amount due to increased influx of Congo moisture. We could very well be seeing the same thing in this GoG record.*

- Regarding moisture sources presented in Figure 3: If recycled moisture is indeed an important component, and there is ample evidence that it should be, then the deep equatorial Congo Basin could easily be a source of moisture even when $P \gg E$. These maps basically show a giant hole over the central Congo Basin (especially 3b, MAM) which does not seem realistic considering the amount of moisture recycling that takes place there.

It is true that a great deal of moisture is recycled in the Congo Basin, but whether it reaches southern Cameroon depends on the wind direction required to transport the moisture, and also whether this moisture is simply re-precipitated out before leaving the Congo Basin, which is likely the case here in MAM and SON.

- **On a separate but related point, we suggest later in the manuscript that recycling of Sahel-Sahara-moisture was likely greater during AHP (lines 227-229).**

20) *Changes in dDwax at 5.5ka are abrupt.*

To the eye, the change in dDwax between 5-6 ka does seem abrupt, and I appreciate the authors' attempt to quantify the rate of change rather than just saying it looks that way. However, I am skeptical of the Rate of Change analysis, particularly as it pertains to the mid-Holocene. In the Supplement the authors admit that the resolution of the dDwax record is too low in the late Holocene (after 5ka) to assess the end of the AHP termination. It is difficult to tell the exact mid-Holocene resolution from Figure 2 as the individual data points are not clearly shown. However the authors give a ~0.2kyr resolution in the caption for Figure 2. If the resolution is 0.2kyr, the low resolution of the record begins around 5kyr, and rates of change are determined by a 5-point smooth, then how trustworthy is it that "the maximum rate of change during the Holocene was at 5.3ka" (lines 120-121)? Couldn't this change at 5.3ka just reflect a change in the resolution of the record after 5ka, rather than an actual increase in dDwax RoC? Since the authors have pitched this paper as an answer to the "abrupt versus gradual" question, this point needs to be crystal clear. I suggest testing this "abrupt termination" assertion with additional analyses, either by increasing the resolution of the record in the late Holocene, or by testing the effects of

resolution using synthetic datasets and the RoC code (or both). In addition, caveats should be explicitly stated in the main text.

- **We have added 18 additional δD_{wax} datapoints to the late Holocene after 7 ka. The average resolution for the period 0 to 5 ka is now 0.16 ka, only slightly higher than the earlier Holocene (5 - 11 ka is 0.13 ka). With the additional data, the dataset is now of high enough resolution for the SiZer analysis and use of the Ruppert-Sheather-Wand bandwidth: it displays a significant decrease between 5.8 and 4.7 ka (Fig. 4). We have increased the size of the data points in Fig. 2.**

21) Changes in dD_{wax} at 5.5ka are of large magnitude (comparable to the YH and H1, both believed to have been driven by NH temperature)

- The SiZer method needs to be described in the main text, not in the Supplement, as it is key to the authors' interpretation of "large and significant" changes in dD_{wax} .

- **We now describe the SiZer method in the main text (lines 109-112) and show the results in Fig. 4. To assess the rapidity, rather than quoting the maximum rate of change we now calculate the mean rate of change for the period of significant change from the Ruppert-Sheather-Wand smooth (line 112-113).**

- The authors argue that the rate of change at 5.5ka is large, but from Figure S3 it appears that this large change only barely crosses the Ruppert-Sheather-Wand bandwidth.

By large we were referring to the 'magnitude' i.e. the rate of change multiplied by the duration of the period of significant change.

- **We no longer to refer to the magnitude and only compare the rate of change and duration of the periods of significant change.**

22) - There seems to be some confusion in the paper about the designation of "convective precipitation." The authors take care to point out "convective" precipitation, for example lines 61 and 107. It's unclear whether the authors just mean precipitation in general, or if they truly only mean convective precipitation (rather than stratiform, which is also present in central Africa for example in Mesoscale Convective Systems). If they mean convective precipitation, are they referring to deep or shallow? These distinctions are important when discussing the mechanisms for windspeeds in the TEJ and AEJ influencing precipitation. If the authors simply mean precipitation in general then they should remove these references to convection, for the sake of the atmospheric convection researchers who read Nat Comms.

We were referring to precipitation in general.

- **As mentioned above, we have removed the references to convection and simply refer to precipitation.**

23) Line 77: The age model is not in fact 'published' but is just given in the Supplement (unless there is a missing reference in the Supplement). Providing the age model in the Supplement is OK, but in the main text please give the basics: What was used to date the core, how many radiocarbon dates, the general age uncertainty, and the age uncertainty around the 5.5ka event in particular.

The references were originally in the caption of Figure S1 to reduce the references in the main text.

- **We have now added the references to the main text and included the number of radiocarbon dates and the mean uncertainty (lines 62-64). We also describe the uncertainty around the duration of the AHP termination (line 222-223).**

24) Line 83: Why are these C3 and C4 end members chosen, rather than using the more updated compilation in Sachse et al. 2012?

This is African vegetation and thus likely more appropriate for our study than the global compilation of Sachse et al 2012.

➤ **We emphasise that these are African plants (lines 115, 117)**

25) Line 88: Please provide the $d^{13}C_{c29}$ values that you cite as evidence for C4-dominated vegetation in the Sahara-Sahel. Also, for the sake of comparability, I strongly recommend using real measurements of plants or sediments rather than the model/reanalysis dataset cited here.

Modern surface sediments from the coast of the Sahel have relatively high $\delta^{13}C$ values due to C₄ grass input. It is however, likely a mixed input, and so not as high as $\delta^{13}C$ values from C₄ plants.

➤ **We refer to Collins et al., 2011 (line 123).**

26) Lines 214-219: The authors might consider CO₂ as a possible forcing, rather than, or in addition to, precession. Nonlinear responses to CO₂ forcing are very plausible.

Agreed, although atmospheric CO₂ increases around 7ka to present. If we assume CO₂ forcing goes the same direction as during the deglacial, this would not explain the aridification at 5.8-4.7 ka.

Reviewer #3

This paper is well motivated and suggests an interesting additional possible mechanism to explain the abrupt AHP termination. One of the main questions I have is whether the paleoproxy is representative of the West African monsoon and Sahel vegetation. Following justification of this point, and some others below, I would consider recommending this manuscript for publication. Major comments:

27). *This study uses a marine core located in the Gulf of Guinea. It appears that the primary sources of terrestrial material to the core are from equatorial Africa. It is not clear how a core with these sources would represent the monsoon and Sahelian vegetation at ~15N, which can be driven by different physical mechanisms than equatorial precipitation. A clearer/stronger justification for the use of this paleo-proxy cite is needed in this regard.*

This may have been unclear in the original version. Indeed the waxes themselves originate from the equatorial Africa, but the incorporated hydrogen isotopic signal integrates over a larger area. In our case, precipitation in Cameroon incorporates moisture, and thus the isotopic signal, from the Sahel-Sahara (see comments 13-15).

➤ **The above point is emphasised (lines 156-157).**

28). *Previous studies have suggested that vegetation-atmosphere feedbacks could have served as a mechanism for the abrupt AHP termination. This study puts forth another possible explanation -- that the initial precipitation decrease leading out of the AHP was driven by northern high-latitude SST cooling that resulted in a teleconnected TEJ response. Some of the wording is too strong given the lack of direct modeling evidence (e.g., line 243: "likely"). I am not necessarily recommending adding a climate modeling component to the study (although it would be interesting!), but the wording should be softened ("could" or "possible") unless climate model experiments are added.*

➤ **We have added in an additional modelling component to the study (see also comments 5 and 7; lines 319-349; Fig. 7). This shows the effect of the north Atlantic cooling on temperature in Africa, which in turn reduces the**

speed of the TEJ.

29) *In addition, I suggest balancing the abstract by including a mention of the role of vegetation-atmosphere feedbacks, as it remains unclear whether the abrupt termination was driven by highlatitude cooling, vegetation feedbacks, or both.*

Good point - this issue was also raised by reviewer two. We suggest that the cooling was likely the trigger for the AHP termination, but that vegetation feedbacks were also active, enhancing the aridification.

➤ **This has been clarified (lines 32-34, 270-271, 360-363).**

30) *Furthermore, paleo-proxy data suggest the onset of the AHP was also abrupt (e.g., deMenocal et al. 2000). Does the highlatitude SST mechanism explain the abrupt AHP onset as well?*

Yes, partly - high latitude SST plays an important role during the deglacial as shown in the model study of Otto-Bliesener et al., (2014). However, in detail the mechanism during the mid-Holocene differs from that during the deglacial. For example the SST changes are much smaller during the mid-Holocene compared to the deglacial. This would also suggest the need for vegetation feedbacks during the mid-Holocene.

➤ **The above points are now described (lines 40-42, lines 360-363)**

31). *Line 47: Some paleo records show an abrupt AHP onset/termination, while others don't. Is there a reason to believe one conclusion vs. the other? Are the records that show a gradual AHP termination anomalies among the greater set? A few words to put this in perspective would help.*

This is an interesting point that was also raised by reviewer 2. An important point is that the abruptness depends on the proxy, (as mentioned earlier with respect to the dust). δD_{wax} reflects atmospheric circulation and a rapid change there indicates a switch in the atmospheric system rather than other processes. In terms of number of records, it is fair to say that records interpreted as gradual are more numerous. However, this includes many records of lower resolution, with poorer chronologies or lower proxy fidelity, and these may be some reasons why the changes do not seem abrupt.

➤ **We now say that 'many' records show a gradual response (line 50). We now discuss several of the records and compilations showing gradual responses (lines 201-208; see also comments 10-12).**

32). *Line 48: Similar comment regarding the coupled models. Are there modeling simulations that suggest that biogeophysical feedbacks could have caused the vegetation collapse? Two studies come to mind - one that simulates an abrupt AHP termination, and another that simulates a gradual one (Claussen et al, 1999; Wang et al. 2005). There may be others as well.*

Indeed, the intermediate complexity models show mixed responses: some show an abrupt response, other do not. Interestingly, most fully coupled GCMs struggle to achieve wet enough conditions during the AHP (e.g. Tierney et al., 2017).

➤ **We have now included the Wang reference (Claussen et al., 1999 was in the original) and state in the introduction that many models have difficulty in simulating wet enough conditions during the introduction (lines 53-55), including references to Zheng and Braconnot, 2013 and Tierney et al., 2017.**

Reviewers' comments:

Reviewer #1 (Remarks to the Author):

This is my second time to review the manuscript. Overall, all my comments are addressed and I am satisfied with the revision. I only have a minor comment.

In the method, the authors should clarify that the hosing experiment was the easiest way to cool the northern hemisphere and should mention that hosing may not have happened.

Reviewer #2 (Remarks to the Author):

Collins et al. have done an excellent, thorough job revising this manuscript. They have added additional measurements, added new model analyses, increased the depth of discussion on key points such as moisture source and interpretation of dD_p , and clarified the uncertainties involved in the abruptness of the mid-Holocene change in their dD_{wax} record. The increased resolution of the record during the AHP termination helps solidify the argument about the rates of change during that time. It will be up to the community to decide whether the high/mid-latitude teleconnections to the TEJ is a plausible way to tip the system toward an arid state, but at least now in the ms it is well-argued and clear, and forms a robust hypothesis that can be tested with future modeling studies.

I only have one major comment, which is below. My other comments follow. All together these are fairly minor revisions. I recommend publication of this manuscript after my comments are addressed.

Major comment:

Figure 5 and Lines 220-256: I support the focus on other leaf wax dD records for the sake of comparability to the GoG record. However it seems odd to only include the Gulf of Aden in Figure 5 and in the discussion when there are several additional high-resolution additional records from the Sahel-Sahara-influenced (and TEJ and AEJ-influenced) parts of tropical Africa. The Lake Tana record should certainly be shown, and possibly the Tanganyika and Victoria records since they are so important to the discussion of the AHP termination as recorded by precipitation dD . In order to show these records of course the authors will have to acknowledge that the timing of terminations in these other records is not the same as the GoG or Gulf of Aden (though importantly, they are all fairly abrupt). Staggered timing of AHP termination is not surprising considering the mixture of moisture sources in tropical Africa, but this must be addressed in order for the arguments in this paper to be robust. Otherwise, the authors are simply cherry-picking the one record that looks the most like their own, which is misleading, and does a real injustice to the rest of the excellent science presented in this paper.

Minor comments:

Line 55: "magnitude of the mid-Holocene precipitation increase" makes it sound like precipitation increased from the mid-Holocene to the present. Suggest changing this to "the intensity of precipitation during the AHP" or similar.

Figure 1: I believe the white box representing leaf wax source area in Figure 2b caption is actually a black box in Figure 1a, this should be clarified. Also, include the yellow dots in the figure caption (also explain why there are 2 yellow dots for Douala). I also recommend making all the text in figures 1b and 1c bigger and thicker so that it is easier to read.

Lines 87-94 and Figure 2: How were the percent contributions to Cameroon precipitation calculated? It seems odd that they would add up to 100%, since the Sahel-Sahara and Gulf of Guinea boxed areas do not cover the entire shaded region in Fig 2a-d, for example the shaded regions in central Africa (around and South of the Congo). Perhaps these are minor contributions but the %Sahel-Sahara and %GoG still shouldn't add up to 100%. Some clarification is needed in the text.

Lines 159-161: I don't understand why the similarity between Douala and N'djamena long-term mean dDp supports the conjecture that dDp in Cameroon is affected by the Sahel-Sahara? GNIP stations in East Africa, southern Africa, etc can also have long-term mean dDp of around -15 or -20 permille but that doesn't mean they have the same moisture sources as Cameroon...

Line 165: Basic context about the iLOVECLIM simulation should be provided here in the main text. What time periods were simulated at what resolution using what forcings? The reader needs to know these details without having to chase down the Supplement and the references cited therein.

Lines 358-365: I recommend making the language in this concluding paragraph more specific. For example, rather than saying that high-and mid-latitude temperature changes "played a decisive role during the mid-Holocene," specify what role exactly. Instead of saying high-latitude temperature changes "resulted" in "large and rapid hydrological change" "in combination" with other feedbacks, be specific: You're talking about the strength of JJA precipitation as it relates to the TEJ and the AEJ, and invoking these high-mid latitude teleconnections to propose a potential tipping point to the system. All of these details can be found elsewhere in the manuscript if read carefully, but this final paragraph is an important chance to summarize the key aspects of the argument, which facilitates the application of this paper to other scientific questions and time periods.

Reviewer #3

The authors have clearly put substantial thought and effort into addressing the reviewer comments. Many of my comments have been fully addressed. Some open points (in blue) remain on a few.

27). This study uses a marine core located in the Gulf of Guinea. It appears that the primary sources of terrestrial material to the core are from equatorial Africa. It is not clear how a core with these sources would represent the monsoon and Sahelian vegetation at ~15N, which can be driven by different physical mechanisms than equatorial precipitation. A clearer/stronger justification for the use of this paleo-proxy cite is needed in this regard.

This may have been unclear in the original version. Indeed the waxes themselves originate from the equatorial Africa, but the incorporated hydrogen isotopic signal integrates over a larger area. In our case, precipitation in Cameroon incorporates moisture, and thus the isotopic signal, from the Sahel-Sahara (see comments 13-15).

→ **The above point is emphasised (lines 156-157).**

The explanation is not yet convincing. It is difficult to see how the waxes would be a suitable proxy for Sahel conditions if only 30% of moisture comes from that region annually. What are the FLEXPART-based estimates during the monsoon peak in JAS? Furthermore, the West African monsoon is known to demonstrate a north-south dipole anomaly, meaning at times when the Sahel is drier than average, the Guinean Coast is usually wetter than average; this further confounds the use of this proxy.

I may be missing something, as isotopic signatures are outside of my expertise, however, the explanation should be understandable by a broad audience for publication in Nature Communications.

28). Previous studies have suggested that vegetation-atmosphere feedbacks could have served as a mechanism for the abrupt AHP termination. This study puts forth another possible explanation -- that the initial precipitation decrease leading out of the AHP was driven by northern high-latitude SST cooling that resulted in a teleconnected TEJ response. Some of the wording is too strong given the lack of direct modeling evidence (e.g., line 243: "likely"). I am not necessarily recommending adding a climate modeling component to the study (although it would be interesting!), but the wording should be softened ("could" or "possible") unless climate model experiments are added.

→ **We have added in an additional modelling component to the study (see also comments 5 and 7; lines 319-349; Fig. 7). This shows the effect of the north Atlantic cooling on temperature in Africa, which in turn reduces the speed of the TEJ.**

The new climate model simulations provide convincing evidence that high-latitude SST cooling can drive a teleconnected response in the West African monsoon system, with a deeper thermal low and weaker TEJ and AEJ all contributing to Sahelian precipitation enhancements. I suggest mentioning the AEJ with the TEJ in the abstract, as both are important circulation changes.

30) Furthermore, paleo-proxy data suggest the onset of the AHP was also abrupt (e.g., deMenocal et al. 2000). Does the highlatitude SST mechanism explain the abrupt AHP onset as well?

Yes, partly - high latitude SST plays an important role during the deglacial as shown in the model study of Otto-Bliesener et al., (2014). However, in detail the mechanism during the mid-Holocene differs from that during the deglacial. For example the SST changes are much smaller during the mid-Holocene compared to the deglacial. This would also suggest the need for vegetation feedbacks during the mid-Holocene.

→ The above points are now described (lines 40-42, lines 360-363)

This is an interesting discussion, but doesn't directly address the point I was attempting to make. The reason I asked whether the high-latitude SST mechanism can explain the abrupt AHP onset as well, is because this can give hints to the leading mechanism(s) that explain the AHP precipitation behavior. If the SST changes were symmetric at the abrupt onset and termination of the AHP, this strongly suggests the SST mechanism is primarily driving the precipitation response. However, if the high-latitude SST cooling at the AHP termination was not matched by a similar high-latitude SST warming at AHP onset, this suggests the SST mechanism played a more secondary role -- in which case, perhaps land-atmosphere interactions were the primary driver, and models are simply deficient in representing this. The manuscript should include discussion along these lines, and the strength of wording regarding the SST mechanism conclusions should reflect this accordingly.

We would like to thank the reviewers for their additional insightful comments and suggestions. We have now modified the manuscript accordingly. Below we address these comments in **green** and describe changes made in **bold**.

Reviewer #1 (Remarks to the Author):

This is my second time to review the manuscript. Overall, all my comments are addressed and I am satisfied with the revision. I only have a minor comment.

1). In the method, the authors should clarify that the hosing experiment was the easiest way to cool the northern hemisphere and should mention that hosing may not have happened.

We have now included this in the manuscript. We feel it is an important aspect to bear in mind and thus that it fits better in the 'results' section (lines 345-346).

Reviewer #2 (Remarks to the Author):

Collins et al. have done an excellent, thorough job revising this manuscript. They have added additional measurements, added new model analyses, increased the depth of discussion on key points such as moisture source and interpretation of dDp, and clarified the uncertainties involved in the abruptness of the mid-Holocene change in their dDwax record. The increased resolution of the record during the AHP termination helps solidify the argument about the rates of change during that time. It will be up to the community to decide whether the high/mid-latitude teleconnections to the TEJ is a plausible way to tip the system toward an arid state, but at least now in the ms it is well-argued and clear, and forms a robust hypothesis that can be tested with future modeling studies.

I only have one major comment, which is below. My other comments follow. All together these are fairly minor revisions. I recommend publication of this manuscript after my comments are addressed.

Major comment:

2). Figure 5 and Lines 220-256: I support the focus on other leaf wax dD records for the sake of comparability to the GoG record. However it seems odd to only include the Gulf of Aden in Figure 5 and in the discussion when there are several additional high-resolution additional records from the Sahel-Sahara-influenced (and TEJ and AEJ-influenced) parts of tropical Africa. The Lake Tana record should certainly be shown, and possibly the Tanganyika and Victoria records since they are so important to the discussion of the AHP termination as recorded by precipitation dD. In order to show these records of course the authors will have to acknowledge that the timing of terminations in these other records is not the same as the GoG or Gulf of Aden (though importantly, they are all fairly abrupt). Staggered timing of AHP termination is not surprising considering the

mixture of moisture sources in tropical Africa, but this must be addressed in order for the arguments in this paper to be robust. Otherwise, the authors are simply cherry-picking the one record that looks the most like their own, which is misleading, and does a real injustice to the rest of the excellent science presented in this paper.

Lake Tana δD_{wax} displays a 60 per mil enrichment at 8ka, much larger than in most other East African records, and was attributed to a decrease in the supply of Congo-derived recycled moisture (Costa et al., 2014). There is, however, little change in δD_{wax} at 5.5 ka at Lake Tana, although there is a decrease in sediment Ti content at about 5 ka, suggesting aridification (Marshall et al., 2011). Similarly, an increase in sedimentary K content at about 5 ka at Chew Bahir (Foerster et al., 2015), and lake level records from Lake Abhe and Lake Ziway Shalla all suggest drying around 5 ka (Gasse and van Campo 1994, Gillespie et al., 1983; Chalieu and Gasse 2002). Thus, it would seem that in the case of Lake Tana, δD_{wax} is strongly influenced by moisture source rather than precipitation amount, perhaps because of its position close to the CAB, and/or its relatively high altitude (1800m). Nonetheless, this remains important to address, when interpreting our data.

In Fig. 5 we now include the δD_{wax} records of Lake Tana, Lake Victoria, Lake Tanganyika, as suggested by the reviewer. Since we previously included discussion of Lake Bosumtwi, we now also include this in Figure 5 for completeness (please note that we are awaiting data for the Lake Bosumtwi confidence intervals from Tim Shanahan - these are not available online. These will be added in before final submission of high quality versions of the figures). We have now incorporated the above records in the 'results' section (lines 218-233) and have re-arranged the order of the 'results' for improved clarity: we firstly discuss each of the δD_{wax} records from Fig. 5 (lines 206-249), including the potential imprint of moisture source variability. We subsequently discuss evidence from other proxies and lake levels (lines 251-274), which do provide support for aridification at around 5.5 ka (lines 270-274).

Minor comments:

3). Line 55: “magnitude of the mid-Holocene precipitation increase” makes it sound like precipitation increased from the mid-Holocene to the present. Suggest changing this to “the intensity of precipitation during the AHP” or similar.

Agreed - this has been changed accordingly.

4). Figure 1: I believe the white box representing leaf wax source area in Figure 2b caption is actually a black box in Figure 1a, this should be clarified. Also, include the yellow dots in the figure caption (also explain why there are 2 yellow dots for Douala). I also recommend making all the text in figures 1b and 1c bigger and thicker so that it is easier to read.

We only referred to the leaf-wax source area in Fig. 2 to avoid overloading Fig. 1

with too many boxes. The black box in Fig. 1a marks the outline of the inset - **this is now stated in the caption**. The second yellow dot close to Douala represents Lake Ossa (**also now in the caption**). **The text on Fig. 1 has been enlarged.**

5). Lines 87-94 and Figure 2: How were the percent contributions to Cameroon precipitation calculated? It seems odd that they would add up to 100%, since the Sahel-Sahara and Gulf of Guinea boxed areas do not cover the entire shaded region in Fig 2a-d, for example the shaded regions in central Africa (around and South of the Congo). Perhaps these are minor contributions but the %Sahel-Sahara and %GoG still shouldn't add up to 100%. Some clarification is needed in the text.

Agreed - this was unclear in the previous version. The reviewer is correct that the two moisture sources would not add up to 100% of that delivered to southern Cameroon, although the contribution from the Congo is relatively minor.

We have now re-phrased the text, quoting in the text the FLEXPART-estimated precipitation amount (mm) derived from the two main sources from Fig. 2 (lines 94-97)

6). Lines 159-161: I don't understand why the similarity between Douala and N'djamena long-term mean δD_p supports the conjecture that δD_p in Cameroon is affected by the Sahel-Sahara? GNIP stations in East Africa, southern Africa, etc can also have long-term mean δD_p of around -15 or -20 permille but that doesn't mean they have the same moisture sources as Cameroon...

Agreed - this may have been confusing. This similarity would, nonetheless, suggest that the moisture sources are not completely different. Additionally, GNIP data also show that there is a relatively weak amount effect in equatorial regions and a stronger amount effect in the Sahel. Given that Cameroon receives moisture from the Sahel, this would suggest that Sahelian precipitation amount change likely has a large imprint on Cameroon δD_p .

We have now removed the Douala - N'djamena comparison but elaborate further on the differences in the amount effect between the two regions (lines 163-176). We include a new figure (Supplementary Fig. 4) to illustrate the amount effect in the two regions. For more details please also see comment 9 below.

7). Line 165: Basic context about the iLOVECLIM simulation should be provided here in the main text. What time periods were simulated at what resolution using what forcings? The reader needs to know these details without having to chase down the Supplement and the references cited therein.

We felt that this is better included in the Methods along with the CCSM3 description, rather than in the main text, so that it is clear to the reader that we have used both CCSM3 and iLOVECLIM for different purposes.

Most of the context of the iLOVECLIM model is now included in the Methods (lines 465-472), and has been removed from the Supplement. We do mention the length of the simulation in the main text because of it's

importance (lines 178-179).

8). Lines 358-365: I recommend making the language in this concluding paragraph more specific. For example, rather than saying that high-and mid-latitude temperature changes “played a decisive role during the mid-Holocene,” specify what role exactly. Instead of saying high-latitude temperature changes “resulted” in “large and rapid hydrological change” “in combination” with other feedbacks, be specific: You’re talking about the strength of JJA precipitation as it relates to the TEJ and the AEJ, and invoking these high-mid latitude teleconnections to propose a potential tipping point to the system. All of these details can be found elsewhere in the manuscript if read carefully, but this final paragraph is an important chance to summarize the key aspects of the argument, which facilitates the application of this paper to other scientific questions and time periods.

We have modified the final paragraph to include the above suggestions (lines 391-396).

Reviewer #3 (Remarks to the Author):

Reviewer #3

The authors have clearly put substantial thought and effort into addressing the reviewer comments. Many of my comments have been fully addressed. Some open points (in blue) remain on a few.

9). This study uses a marine core located in the Gulf of Guinea. It appears that the primary sources of terrestrial material to the core are from equatorial Africa. It is not clear how a core with these sources would represent the monsoon and Sahelian vegetation at ~15N, which can be driven by different physical mechanisms than equatorial precipitation. A clearer/stronger justification for the use of this paleo-proxy cite is needed in this regard.

This may have been unclear in the original version. Indeed the waxes themselves originate from the equatorial Africa, but the incorporated hydrogen isotopic signal integrates over a larger area. In our case, precipitation in Cameroon incorporates moisture, and thus the isotopic signal, from the Sahel-Sahara (see comments 13-15).

→ The above point is emphasised (lines 156-157).

The explanation is not yet convincing. It is difficult to see how the waxes would be a suitable proxy for Sahel conditions if only 30% of moisture comes from that region annually. What are the FLEXPART-based estimates during the monsoon peak in JAS? Furthermore, the West African monsoon is known to demonstrate a north-south dipole anomaly, meaning at times when the Sahel is drier than average, the Guinean Coast is usually wetter than average; this further confounds the use of this proxy.

I may be missing something, as isotopic signatures are outside of my expertise, however, the explanation should be understandable by a broad audience for publication in Nature Communications.

During the southern Cameroon wet season (SON) the Sahel-Sahara contributed 266mm of precipitation and the Gulf of Guinea (now more precisely termed

southeast Atlantic) contributed 438mm (thus, in a ratio of about 2:3). This is slightly higher than mean annual: the Sahel-Sahara and southeast Atlantic contributed 568mm and 1492mm of precipitation (a ratio of 3:7). Importantly, however, as mentioned in comment 6 above, the 'amount effect' (the negative correlation between δD_p and amount) is stronger in the Sahel, i.e. on interannual timescales mean-annual precipitation amount and δD_p values from GNIP stations display a steeper slope and a greater magnitude of δD_p variability (as is now shown in Supplementary Fig. 4). This would suggest that precipitation changes in the Sahel would likely contribute an isotopic signal of greater magnitude than precipitation changes in southern Cameroon. Thus, even if the Sahel only contributes around 30-40% of the moisture to southern Cameroon, one would expect the imprint of the Sahelian signal to be relatively strong in Cameroon. Thus, Cameroon δD_p likely represents an integrated precipitation of Cameroon and the central Sahel-Sahara. In support of this, the iLOVECLIM simulations also suggest that Cameroon δD_p integrates Sahel-Sahara precipitation. **We now include discussion of the above (lines 163-176) and include a new figure (Supplementary Fig. 4).**

Indeed the West African monsoon clearly demonstrates a north-south dipole on interannual timescales (e.g. Cook and Vizy 2006), and this is often associated with Gulf of Guinea SSTs (which, we note, would not explain the AHP termination; Fig. S6). It is not clear, however, that the dipole mechanism would explain the hydrological changes at the AHP termination. For example, we know from many proxies (e.g. Shanahan et al., 2015) that the Sahel-Sahara was wetter during the mid-Holocene. However, proxy records from the Gulf of Guinea (eg Weldeab et al., 2008) from Barombi Mbo (Giresse et al 1994) and lake levels from Bosumtwi (Shanahan et al., 2015) indicate southern Cameroon and the equatorial regions were also wetter in the mid-Holocene, which would not be in line with the dipole mechanism.

10). Previous studies have suggested that vegetation-atmosphere feedbacks could have served as a mechanism for the abrupt AHP termination. This study puts forth another possible explanation -- that the initial precipitation decrease leading out of the AHP was driven by northern high latitude SST cooling that resulted in a teleconnected TEJ response. Some of the wording is too strong given the lack of direct modeling evidence (e.g., line 243: "likely"). I am not necessarily recommending adding a climate modeling component to the study (although it would be interesting!), but the wording should be softened ("could" or "possible") unless climate model experiments are added. \rightarrow We have added in an additional modelling component to the study (see also comments 5 and 7; lines 319-349; Fig. 7). This shows the effect of the north Atlantic cooling on temperature in Africa, which in turn reduces the speed of the TEJ.

The new climate model simulations provide convincing evidence that high-latitude SST cooling can drive a teleconnected response in the West African monsoon system, with a deeper thermal low and weaker TEJ and AEJ all contributing to Sahelian precipitation enhancements. I suggest mentioning the AEJ with the TEJ in the abstract, as both are important circulation changes.

We have added the AEJ to the abstract.

11) Furthermore, paleo-proxy data suggest the onset of the AHP was also abrupt (e.g., deMenocal et al. 2000). Does the high-latitude SST mechanism explain the abrupt AHP onset as well? Yes, partly - high latitude SST plays an important role during the deglacial as shown in the model study of Otto-Bliesener et al., (2014). However, in detail the mechanism during the mid-Holocene differs from that during the deglacial. For example the SST changes are much smaller during the mid-Holocene compared to the deglacial. This would suggest the need for vegetation feedbacks during the mid-Holocene.

→ The above points are now described (lines 40-42, lines 360-363)

This is an interesting discussion, but doesn't directly address the point I was attempting to make. The reason I asked whether the high-latitude SST mechanism can explain the abrupt AHP onset as well, is because this can give hints to the leading mechanism(s) that explain the AHP precipitation behavior. If the SST changes were symmetric at the abrupt onset and termination of the AHP, this strongly suggests the SST mechanism is primarily driving the precipitation response. However, if the high-latitude SST cooling at the AHP termination was not matched by a similar high-latitude SST warming at AHP onset, this suggests the SST mechanism played a more secondary role -- in which case, perhaps land-atmosphere interactions were the primary driver, and models are simply deficient in representing this. The manuscript should include discussion along these lines, and the strength of wording regarding the SST mechanism conclusions should reflect this accordingly.

Agreed - land-atmosphere interactions ('feedbacks') were likely an important factor in enhancing aridification. Various feedbacks have been proposed (vegetation, soil moisture, dust, lakes and wetlands), although the latest models suggest that they were unlikely to have acted alone in tipping the climate towards an arid state. Indeed, this would suggest that either the models are deficient, or that there was additional factor involved in triggering the AHP termination. Given the evidence for (albeit small) cooling at 5.5 ka and the modelled effect that this has on the TEJ, this led us to suggest that the teleconnection with the TEJ was the missing factor. Nonetheless, as the reviewer points out, the magnitude of high-latitude SST change is asymmetrical at the onset and termination of the AHP: the magnitude of high-latitude SST changes is, in fact, larger at the onset than at the termination. However, at the AHP onset, high-latitude SST change was accompanied by changes in the tropics, not evident in the termination (Supplementary Fig. 6), and thus one might argue that the high to low latitude SST gradient is more important for African hydrology. Additionally, however, at the AHP onset, there are a whole range of other global climate changes taking place, such as increasing tropical SST, increasing CO₂, northern hemisphere ice-sheet retreat, sea level rise, which are not present in the mid-Holocene, and very likely affected African hydrology at the AHP onset. Thus, it is difficult to directly compare the onset and termination. Nonetheless, given the range of climate modelling work showing evidence for positive land-atmosphere interactions (positive feedbacks), we agree that high-latitude temperature was unlikely to have acted alone at the AHP, but instead that various feedbacks played a role in enhancing aridification, once it was triggered by high-latitude temperature

decrease.

The role of several land-atmosphere interactions ('feedbacks') is discussed (lines 276-319). We now include the reviewers' point that the apparently small effect of feedbacks in simulations may, potentially, be related to a model deficiency (lines 312-313). In the 'discussion', we now also describe the lack of a one to one response of African precipitation to high latitude temperature, but emphasise that the onset and termination cannot be directly compared (lines 379-385). We state that feedbacks very likely played a role in enhancing aridification (lines 385-387). We also include a summary of this in the concluding paragraph (lines 391-396).